# Barcoding of small extracellular vesicles with CRISPR-gRNA enables comprehensive, subpopulation-specific analysis of their biogenesis and release regulators

Koki Kunitake [1,2], Tadahaya Mizuno [2], Kazuki Hattori [3], Chitose Oneyama [4], Mako Kamiya [5], Sadao Ota [3], Yasuteru Urano [1,2] & Ryosuke Kojima [1,6,7] ✉

Small extracellular vesicles (sEVs) are important intercellular information transmitters in various biological contexts, but their release processes remain poorly understood. Herein, we describe a high-throughput assay platform, CRISPR-assisted individually barcoded sEV-based release regulator (CIBER) screening, for identifying key players in sEV release. CIBER screening employs sEVs barcoded with CRISPR-gRNA through the interaction of gRNA and dead Cas9 fused with an sEV marker. Barcode quantification enables the estimation of the sEV amount released from each cell in a massively parallel manner. Barcoding sEVs with different sEV markers in a CRISPR pooled-screening format allows genome-wide exploration of sEV release regulators in a subpopulation-specific manner, successfully identifying previously unknown sEV release regulators and uncovering the exosomal/ectosomal nature of CD63+/CD9+ sEVs, respectively, as well as the synchronization of CD9+ sEV release with the cell cycle. CIBER should be a valuable tool for detailed studies on the biogenesis, release, and heterogeneity of sEVs.

Membrane-enclosed extracellular vesicles (EVs) released by cells are typically classified into several subgroups depending on their size or origin, including small EVs (sEVs, 30–200 nm in diameter), medium/large EVs (EVs larger than sEVs), exosomes (multivesicular body (MVB) derived), and ectosomes (plasma-membrane derived, also known as microvesicles)[1]. Among them, sEVs containing various biomolecules are important mediators of cell-to-cell communication in both physiological and pathological contexts, including cancer metastasis[2,3]. These facts highlight the potential of sEV biogenesis and release processes ("release" processes hereafter) as therapeutic targets[3,4]. Furthermore, sEVs are also attracting attention as highly biocompatible delivery vesicles[3,5], and therefore methods to control/enhance their production are of great interest for biotechnological applications. Despite the importance of sEV release processes, a comprehensive understanding of their regulation has remained elusive for several reasons[6–9]. Firstly, multiple biological molecules are involved in interconnected pathways, which are difficult to analyze with conventional low-throughput assays using small-molecule inhibitors or siRNAs in separate wells (Fig.1a, upper). Secondly, many of the regulators of sEV release also affect cellular activities, including viability, which hampers high-throughput identification of factors controlling sEV release. Thirdly, sEVs are

[1]Graduate School of Medicine, The University of Tokyo, Tokyo, Japan. [2]Graduate School of Pharmaceutical Sciences, The University of Tokyo, Tokyo, Japan. [3]Research Center for Advanced Science and Technology, The University of Tokyo, Tokyo, Japan. [4]Division of Cancer Cell Regulation, Aichi Cancer Center Research Institute, Nagoya, Japan. [5]Department of Life Science and Technology, Institute of Science Tokyo, Kanagawa, Japan. [6]PRESTO, Japan Science and Technology Agency, Kawaguchi, Saitama, Japan. [7]FOREST, Japan Science and Technology Agency, Kawaguchi, Saitama, Japan. ✉e-mail: kojima@m.u-tokyo.ac.jp

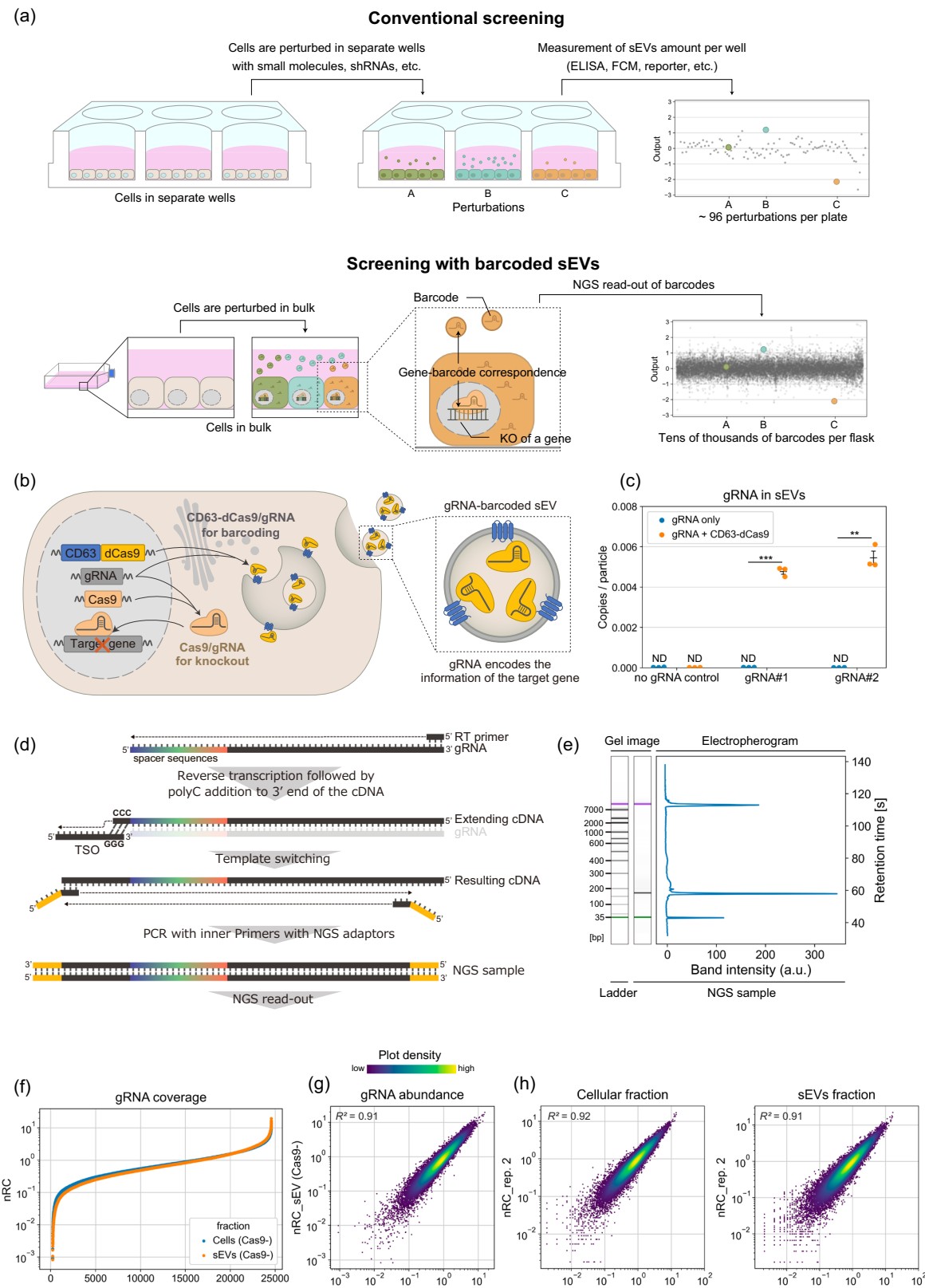

heterogeneous, and the release mechanisms of different sub-populations of sEVs are hard to differentiate.

Here we report a high-throughput pooled screening system that overcomes these limitations and its application to identify key players of sEV release processes. We actively incorporated guide RNA (gRNA) for Cas9 into sEVs through the interaction of gRNA and dead Cas9 (dCas9) fused with an sEV marker in a pooled CRISPR screening format. This allows sEV-loaded gRNA to work as a "barcode" linking each sEV to the perturbation of gene expression in its originating cell. Quantification of the composition of barcode gRNA in both sEVs and cells allows high-throughput, genome-wide exploration of genes involved in sEV release while canceling out the effects on cellular

**Fig. 1 | Concept and creation of gRNA-barcoded sEVs for high-throughput analyses of their release regulators. a** Schematic illustration of the use of "barcoded" sEVs. Conventional biological assays (e.g., by treating cells in separate wells and measuring the amount of sEVs released in each well) offer only rather low throughput. On the other hand, sEVs barcoded with CRISPR gRNA allow for the identification of sEV release regulators in a pooled manner. **b** Schematic illustration of the generation of sEVs barcoded with gRNA. By expressing a fusion protein of CD63 (sEV marker) and dead Cas9 (dCas9) together with Cas9 in the sEV-producing cells, gRNA used for gene knockout can be actively encapsulated in sEVs. **c** gRNA abundance in sEVs isolated from culture media of HEK293T or HEK293T cells expressing CD63-dCas9 stably transduced with gRNA using lentivirus. The copy numbers of gRNAs were determined by qPCR and divided by the particle number. The Ct values of qPCR for gRNA#1 and #2 without CD63-dCas9 were not significantly different from the no-gRNA control (shown as ND: not determined). *p*:

one sample *t*-test under the null hypothesis where copies/particle equals 0. **p < 0.005, ***p < 0.0005. Error bars represent ± SEM of biological replicates (*n* = 3). **d** Schematics of the developed method for spacer amplification for next-generation sequencing (NGS). **e** A Bioanalyzer electropherogram of an NGS sample prepared from sEV RNA as shown in Fig. 1d. **f** Coverage of gRNAs. Cells expressing CD63-dCas9 were transduced with a library of 24,569 gRNAs using lentivirus (DTKP library). RNAs extracted from cells and sEVs released from them were processed for NGS read-out. Read counts for every single gRNA were divided by the total sample count, then multiplied by the number of gRNAs in the library (24,569) to calculate the normalized read count (nRC), such that the mean value becomes 1. Data are shown as mean nRC from two biological replicates. **g** Correlation between nRCs of 24,569 gRNAs in sEVs and cellular fractions from 2 replicate cultures of Cas9⁻ cells. **h** Reproducibility of nRCs of 24,569 gRNAs from cellular and sEVs fractions measured by NGS.

activities (e.g., proliferation, barcode transcription). We call this assay platform CRISPR-assisted individually barcoded sEV-based release regulator (CIBER) screening. CIBER screening using multiple sEV markers in combination with bioinformatic analyses revealed both known and previously unknown factors controlling sEV release processes, uncovering different effects of V-type ATPases, mitochondrial electron transport, and the cell cycle on the release of CD63⁺ and CD9⁺ sEVs. We believe this work provides a basis for detailed studies on the biogenesis, release, and heterogeneity of sEVs. We discuss the potential of this sEV-barcoding platform for various future applications.

## Results

### Concept and design of a library of sEVs barcoded with CRISPR-Cas9 gRNA

In CRISPR pooled screening, Cas9-expressing cells are transduced with a library of gRNAs to perturb gene expression. These genome-incorporated gRNAs in turn enable estimation of the numbers of cells bearing different gene knockouts via read-out of the gRNA by next-generation sequencing (NGS) (Supplementary Fig. 1)[10]. Analogously, we hypothesized that it would be possible to estimate the numbers of sEVs if we could efficiently load sEVs with gRNA transcribed in their originating cells, thereby enabling identification of regulators of sEV release in a pooled, massively parallel manner (Fig.1a, lower). In this setting, each transcribed gRNA will encode information regarding the perturbed gene in its originating cell as a barcode, and quantification of the gRNA barcodes in sEVs by next-generation sequencing (NGS) should allow quantification of sEVs released from each cell.

It has been suggested that stochastic packaging of biomolecules into sEVs is a rather rare event[11]. Therefore, simple overexpression of gRNA inside the cells would be insufficient to create high-quality gRNA-barcoded sEVs. On the other hand, we and several other researchers have reported that specific RNAs can be actively loaded into sEVs by synthetic interaction with an RNA-binding protein (RBP) fused with an sEV marker protein (e.g. CD63)[12–14]. Building on this work, we expected that gRNA could be efficiently loaded into sEVs by using dCas9 as a strong RBP for gRNA (Fig. 1b). We confirmed that CD63-dCas9 fusion protein can be successfully loaded in sEVs without changing the general characteristics of sEVs (according to the MISEV2023 criteria[1]) when it is expressed in the sEV producer cells (Supplementary Fig. 2). When we infected HEK293T cells expressing CD63-dCas9 with lentivirus encoding single gRNAs, approximately 1 copy of gRNA per 200 sEVs was detected by qPCR, while the gRNA in sEVs was nearly undetectable in the absence of CD63-dCas9 (Fig. 1c), strongly supporting the conclusion that CD63-dCas9 can efficiently recruit gRNA into sEVs.

Next, we developed a strategy to amplify gRNA spacers directly from a mixture of transcribed gRNAs (Fig. 1d). Unlike normal CRISPR screening that decodes gRNA spacers in the genome, this approach is essential for the parallel analysis of multiple gRNAs in sEVs by next-

generation sequencing (NGS). Considering that the spacer is located at the 5′-end of gRNA, we adopted SMART technology, which adds a common sequence at the 5′-end during reverse transcription by template switching[15]. By employing nested PCR, we could selectively amplify the gRNA spacers from RNAs in sEV with a common pair of primers (Fig. 1e).

We then infected HEK293T cells expressing CD63-dCas9 with a library of 24,569 gRNAs[16]. With the developed spacer amplification method, we succeeded in detecting about 99% of spacer sequences by NGS with a low skew ratio in both cells and sEVs (Fig. 1f, Supplementary data 1, 2). Importantly, when we compared the abundance of each gRNA spacer in the cellular and sEV fractions, a linear relationship was observed (Fig. 1g), indicating that each gRNA was loaded uniformly into sEVs regardless of its spacer sequence. It is noteworthy that keeping 500 cells/gRNA was sufficient to reproducibly achieve such a high barcoding-decoding performance (Fig. 1h), because this cell number is comparable to that utilized in normal pooled CRISPR screening[10]. We also note that the selection of dCas9 as the strong RBP for gRNA was important because the sEV barcoding efficacy was much lower when a bacteriophage MS2 coat protein was used as the RBP for gRNA bearing MS2-binding motifs[17] (Supplementary Fig. 3). After confirming that the expression of Cas9 in addition to CD63-dCas9 does not change sEV size (Supplementary Fig. 4), that the barcoding performance was consistently high when Cas9 was additionally expressed (Supplementary Fig. 5, Supplementary Data 1, 2), and that the co-expression of CD63-dCas9 does not competitively inhibit gRNA-driven gene knockout in Cas9-expressing cells (Supplementary Fig. 6), we decided to apply this dCas9-based sEV barcoding strategy for downstream screening.

### High-throughput evaluation of the effect of Cas9-induced perturbation on sEVs release

HEK293T cells expressing CD63-dCas9 with/without Cas9 (hereafter called Cas9⁺ and Cas9⁻ cells in this section) were infected with lentivirus encoding CRISPR knockout sub-pool gRNA libraries[16] listed in Supplementary Fig. 7a (covering 10,527 genes in total) and gRNAs were extracted and sequenced from both sEV and cellular fractions (Supplementary Fig. 7b, Supplementary Data 1, 2). Although exploration of sEV release regulators could in principle be performed by comparing the gRNA abundances in sEV fractions from Cas9⁺ and Cas9- cells, it is important to consider that the amount of each gRNA in the sEV fraction would also be affected by the change in the cellular fraction due to the gene knockout (e.g., changes of proliferation rate, barcode transcription, etc.; Fig. 2a). Indeed, when we checked the $\log_2$(fold change (FC)) of the abundance of each gRNA between Cas9⁺ and Cas9- samples in sEVs and cells (hereafter referred to as $FC_{sEVs}$ and $FC_{cells}$, respectively), gRNAs targeting POLR3 subunits (regulating the transcription of gRNAs[18]) generally showed similarly low values for both parameters (Supplementary Fig. 8). Furthermore, when we plotted $FC_{sEVs}$ and $FC_{cells}$ of all gRNAs used, a linear relationship was observed

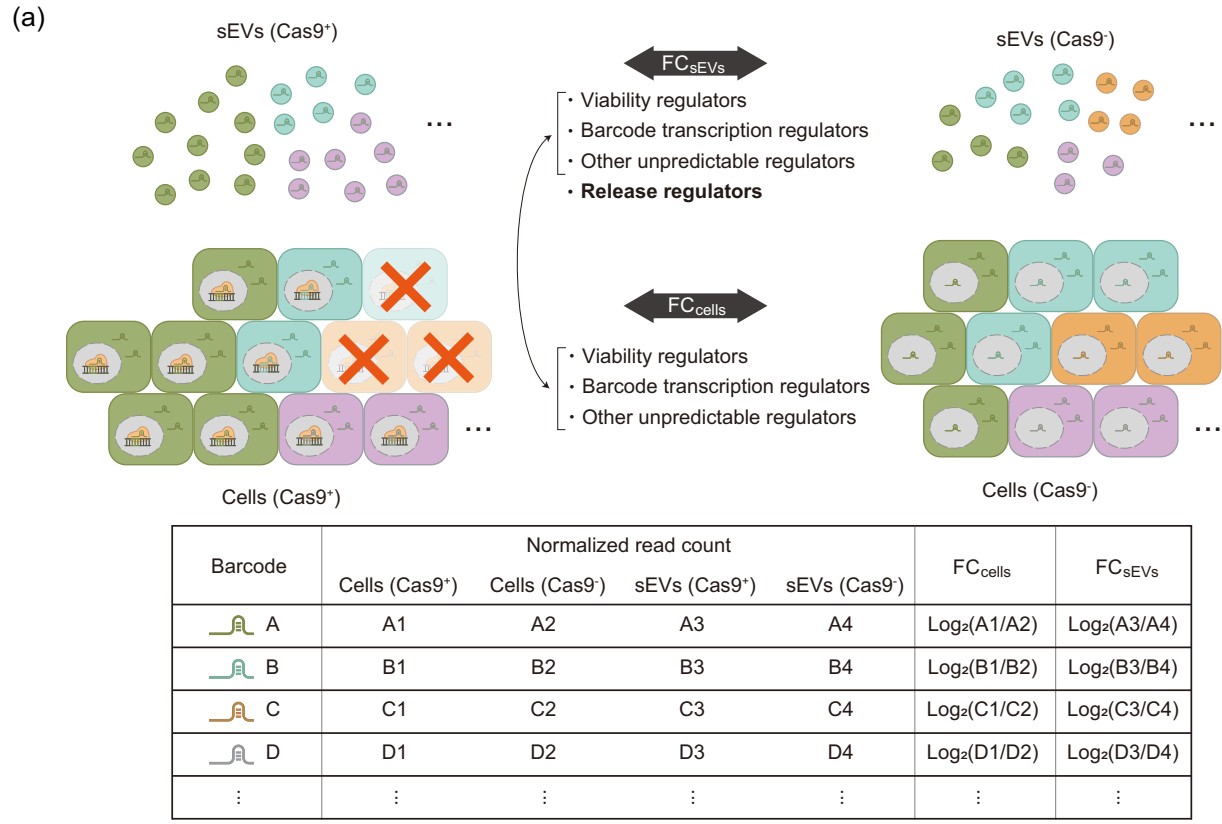

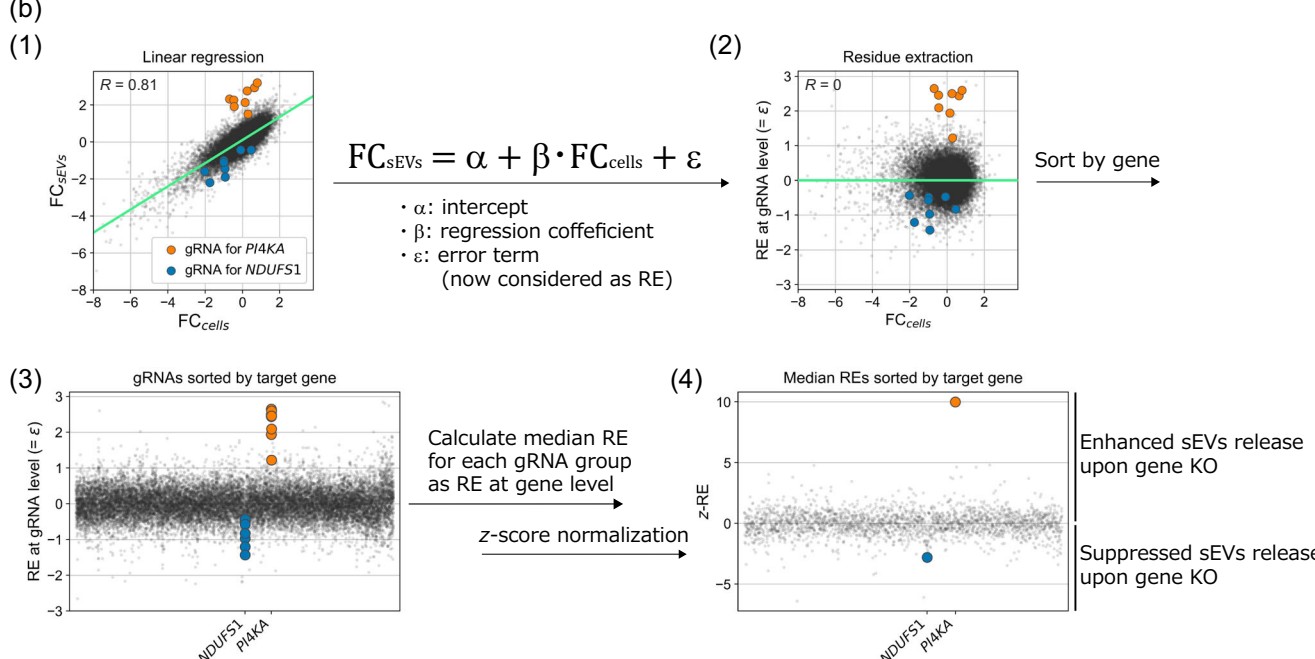

**Fig. 2 | Procedure of CIBER screening. a** HEK293T CD63-dCas9 cells with/without Cas9 are lentivirally transduced with the gRNA libraries, and the RNA is harvested from both cellular and sEV fractions of these cells. After NGS analysis of the gRNA composition in each sample, $FC_{sEVs}$ and $FC_{cells}$ (FC stands for fold change) of each gRNA are calculated as described. The $FC_{cells}$ of each gRNA reflects the change in the abundance of each gRNA in cells due to the KO of viability regulators, barcode transcription regulators, etc. $FC_{sEVs}$ reflects the effect of sEV release regulators in addition to the factors affecting $FC_{cells}$. Therefore, the actual contribution of each gRNA (and gene) to the change of sEV release can be estimated from both $FC_{sEVs}$ and $FC_{cells}$ by means of the following procedure. **b** Calculation of $z$-normalized release effect ($z$-RE) from raw gRNA read counts. (1) Each gRNA was plotted with $FC_{sEVs}$ on the $y$-axis and $FC_{cells}$ on the $x$-axis. A linear relationship between $FC_{cells}$ and $FC_{sEVs}$ is observed. The regression line is shown in green (the results for the DTKP library, consisting of 24,569 gRNAs (see Supplementary Fig. 7), are shown: $R = 0.81$). For clarity, gRNAs targeting *PI4KA* and *NDUFS1*, which were subsequently validated as true hits, are highlighted in different colors. (2) Residue values of each gRNA obtained by performing linear regression on (1) are displayed as RE at the gRNA level. (3) gRNAs are sorted according to their target genes. (4) $z$-RE of each gene is calculated by taking the median value of RE at the gRNA level.

(Fig. 2b(1), Supplementary Data 3). These facts indicate that the change of gRNA level in sEVs is highly dependent on the change of cellular gRNA level ($R = 0.81$), and this led us to introduce a score called "release effect" (RE) to find the true sEV release regulators. First, we performed linear regression using $FC_{sEVs}$ and $FC_{cells}$ and defined the 'RE at the gRNA level' as a residue with respect to the regression line. Then, the median RE among gRNAs targeting the same gene was adopted as the RE at the gene level, and the z-normalized RE (z-RE) was used as the indicator of the contribution of each gene to sEV release (Fig. 2b). This screening pipeline was designated as **C**RISPR-assisted **I**ndividually **B**arcoded s**E**V-based release **R**egulator (CIBER) screening, in which the z-REs of genes that enhance/suppress sEVs release upon KO should be larger/smaller than 0, respectively.

## Results of CD63-dCas9-based CIBER screening in HEK293T cells

The results of the CD63-dCas9-based CIBER screening (hereafter CD63-CIBER) in HEK293T cells are shown in Fig. 3a and Supplementary Data 4. The genes showing z-REs of >1.65 and <−1.65 are treated as "upper hits" (231 genes) and "lower hits" (309 genes), respectively (if the tested population follows a normal distribution, the z-scores of 1.65 and −1.65 corresponds to the 95th percentile and the 5th percentile, respectively[19]). We chose these values expecting that the combination of downstream bioinformatic analysis with this relatively weak threshold would enable robust analysis.

Firstly, we should emphasize that *CD63* was detected as one of the top lower hits. This is exactly what we expected, because *CD63*-targeting gRNAs induce KO of CD63-dCas9 as well, which prevents gRNAs from being loaded into sEVs. Motivated by this confirmation that the positive control gene worked, we validated the screening results with an orthogonal assay using a luciferase reporter CD63-nluc that enables the estimation of CD63$^+$ sEV amount by simple luminescence measurement of the cell culture supernatant[12,20]. *FASN* and *PI4KA* were among the top lower/upper hits, respectively, and indeed, treatment of the cells with small-molecule inhibitors against the translated proteins drastically downregulated/upregulated the CD63-nluc signal, respectively (Fig. 3b, c, Supplementary Fig. 9). These effects were also directly observed by nanoparticle tracking analysis (NTA) with native sEVs (Fig. 3d, e, Supplementary Fig. 10), confirming that CIBER screening can identify previously unknown regulators of sEV release. The multiple hit genes highlighted in Fig. 3a were also validated by siRNA-based assays to be the true hits (Fig. 3f, Supplementary Fig. 11). (See also Supplementary Fig. 12 for false a positive gene as well as the selection of the tested genes. 10 out of 14 genes were validated to be true hits.)

Interestingly, we noticed that many of the proteins encoded by the hit genes are among the top 1000 proteins listed in Vesiclepedia[21,22], a database compiling proteins frequently detected in/ on sEVs (Fig. 3g, Supplementary Fig. 13), which implies that many proteins regulating sEV release are enriched in/on sEVs themselves (note that this does NOT mean that CIBER screening identifies only genes encoding sEV-resident proteins). STRING analysis[23] confirmed that the many of the hit genes, including those encoding the sEVs-resident proteins, were functionally connected to the validated genes as well as their neighbor genes and other hit genes previously reported as sEV release regulators, including *VTI1B*[24] and *ATP6V1A*[25] (Fig. 3h). Further, many of the hit genes also have biologically confirmed interactions with sEV release regulators such as *ARF6*[26], *HGS*[27], *PDCD6IP* (ALIX)[28], *RAB5A*[29], *RAB9A*[29], *SMPD3*[30], and SNARE proteins[24] (Supplementary Fig. 14). These results demonstrate that genes regulating sEVs release are enriched in the CIBER screening hits.

We also performed gene ontology (GO) enrichment analysis[31,32] of hit genes to extract putative biological processes relevant to sEVs release (Fig. 3i, Supplementary Data 5). It is noteworthy that the GO terms involved in vesicle-mediated transport are enriched in lower hits in this assay. Multiple previously unknown gene sets, such as those

involved in tRNA metabolism (lower hits) and cholesterol metabolism (upper hits), are suggested to be involved in sEV release. Interestingly, GO terms grouped as "endosome organization", including ESCRT proteins, were detected as strong upper hits in the screening, which we had not expected since they are known to be essential for sEV release[6,7]. However, validation with siRNA for two ESCRT members *PTPN23* and *VPS28* (Fig. 3f) indicated that depletion of ESCRT proteins could indeed upregulate sEV release in our setting, as also reported in a different study[33]. The functions of ESCRT proteins are quite complex[34] and some sEV release is known to be ESCRT-independent[35], so we presume that ESCRT downregulation could act differently on sEV release depending on the experimental setting. In any case, these data support the conclusion that CIBER screening can comprehensively identify multiple genes and gene sets that significantly affect sEV release processes.

Building on these results, we set out to further demonstrate the usefulness of CIBER screening by applying it in other contexts as follows.

## Applicability of CIBER screening in a cancer cell line

It has been suggested that the regulators of sEV release processes could be cell-type-dependent, and therefore that cancer-cell-specific inhibition of sEV release processes might be a promising therapeutic strategy[3,4]. From this perspective, we conducted another CIBER screening with the DTKP library (2333 genes with 24,569 gRNAs, drug targets, kinases, and phosphatases) in SH-SY5Y cells (derived from human neuroblastoma) expressing CD63-dCas9 (Supplementary Fig. 15, Supplementary Data 2, 4). The screening was successful in this cell line, supporting the portability of CIBER screening to other cell lines. *PI4KA* was again detected as one of the top upper hits and was validated as a true hit, suggesting that this gene may serve as a potent sEV release regulator in multiple cell lines (Supplementary Fig. 15a, b). We also focused on *PKM* and *PGK1* as candidate SH-SY5Y-specific lower hits compared to HEK293T cells. The knockdown of these genes indeed decreased the CD63$^+$ sEV release more significantly in SH-SY5Y cells than in HEK293T cells (Supplementary Fig. 15c), confirming the potential of CIBER screening to identify cell-type-specific sEV release regulators.

## Comparison of the regulators of CD63$^+$ and CD9$^+$ sEVs

sEVs are known to be heterogeneous, but how the release of different sEV subpopulations is controlled remains poorly understood[6]. We hypothesized that CIBER screening using different sEV markers to load gRNA would uncover the sEV release regulators in a subpopulation-specific manner.

Focusing on CD9 as another extensively used sEV marker, we conducted additional CIBER screening using CD9-dCas9 (CD9-CIBER hereafter; Supplementary Fig. 16, Supplementary Data 4) in HEK293T cells. Firstly, we confirmed that CD9 was detected as a strong lower hit in the new screening, while CD63 was not (Fig. 4a), establishing the flexibility of our barcoding system to selectively load gRNA into the targeted sEV subpopulation. The z-REs of each gene in each screening are somewhat similar ($R = 0.43$ for all genes (Fig. 4a), 0.63 for ESCRT genes (Supplementary Fig. 17)) and about 30% of the hit genes overlapped (101 genes for lower hits and 64 genes for upper hits, Supplementary Fig. 18a), suggesting that there are many common regulators of CD63$^+$ and CD9$^+$ sEV release in HEK293T cells. *FASN* and *PI4KA* were among the shared lower/upper hits, and the effects of treatment with inhibitors of these proteins on CD63-nluc and CD9-nluc sEVs were confirmed to be similar (Fig. 3b, c, Supplementary Fig. 16c, d). It seems reasonable that multiple genes are common regulators of the release of CD63$^+$ sEVs and CD9$^+$ sEVs, considering that some proportion of sEVs should express both CD63 and CD9.

We next set out to compare the results of CD63-CIBER and CD9-CIBER in HEK293T cells by GO enrichment analysis and

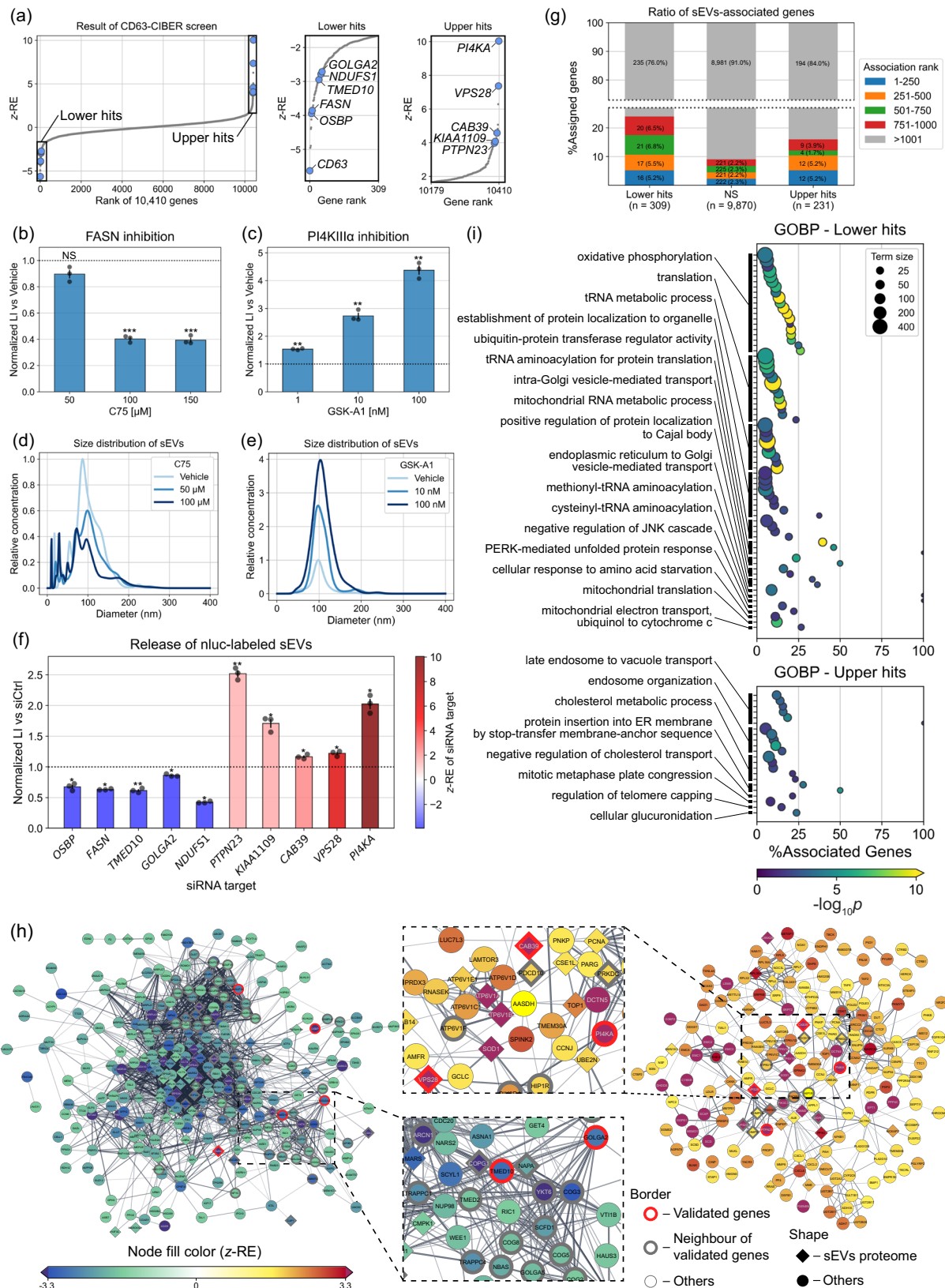

GSEAPreranked, a type of gene set enrichment analysis (GSEA)[36,37], aiming for the robust detection of factors that act differently on the release of different sEV subpopulations. Firstly, we noticed the appearance of significant GO terms grouped as oxidative phosphorylation (OxPhos) only in the lower hits of CD63-CIBER (Fig. 3i); only 2 out of 19 terms were significant for CD9-CIBER and genes annotated to

these terms (mostly encoding mitochondrial proteins) were CD63-CIBER-specific lower hits (Fig. 4b, Supplementary Fig. 18b–e). We decided to pursue this further, because the relationship between sEV heterogeneity and mitochondrial ATP synthesis is not well understood. We conducted a parallel GSEAPreranked by using z-REs of each screening as a pre-ranked query array, and the volcano plot of the

**Fig. 3 | The results of the CD63-dCas9-based CIBER screening. a** $z$-REs of 10,410 genes (left panel). The genes showing $z$-REs larger than 1.65 and lower than −1.65 are considered as "upper hits" and "lower hits", respectively, and are displayed in magnified panels (center and right). Individual $z$-RE values for gRNAs and genes are listed in Supplementary Data 3 and 4. **b**, **c** Validation of the effects of PI4KA and FASN by CD63-nluc reporter assay. HEK293T cells stably expressing CD63-nluc were treated with C75, a FASN inhibitor (**b**) or GSK-A1, a PI4KIIIα (translated from *PI4KA*) inhibitor (**c**). After each treatment, the luminescence from the culture media (CM) was measured. The luminescence intensity of each sample was normalized by the protein amount in the cellular fraction to cancel out the effect of cell viability. Size distribution of sEVs in CM of HEK293T treated with C75 (**d**) or GSK-A1 (**e**), measured by NTA. Each result is the average of 3 sequential measurements of each sample. **f** CD63-nluc reporter assay for hit genes by siRNAs. Cells are transfected with siRNA targeting one of the hit genes and processed as shown in Fig. 3b, c. **g** Percent of sEVs-associated genes for Lower hits, not significant (NS) and Upper hits. sEVs-associated genes were compiled from the Vesiclepedia database. **h** Protein-protein interaction (PPI) network among lower hits (left) and upper hits (right) analyzed by StringApp in Cytoscape. Only nodes connected to at least one other node are shown. The density of the edge is proportional to the strength of the PPI evidence (**i**) Results of gene ontology biological process (GOBP) enrichment analysis. The dot color indicates the -$\log_{10}p$-value by two-tailed Fisher's exact test with Holm correction. Only terms with adjusted *p*-values lower than 0.05 are displayed. Similar terms are grouped and represented by the most significant term in each group (See Methods). Throughout the figure, error bars represent ±SEM of biological replicates ($n = 3$). *p*: two-tailed Welch's *t*-test with Holm correction (see also Supplementary Fig. 12). \*$p < 0.05$, \*\*$p < 0.005$, \*\*\*$p < 0.0005$. NS, not significant.

analyzed ~12,000 gene sets again suggested that inhibition of mitochondrial ATP biosynthesis would more significantly inhibit CD63⁺ sEV release (Fig. 4c, Supplementary Fig. 19, Supplementary Data 6). Indeed, the treatment of HEK293T cells with rotenone (a mitochondrial complex I inhibitor) more strongly downregulated the release of CD63⁺ sEVs than CD9⁺ sEVs; this was consistently confirmed with native sEVs by a sandwich ELISA of CD63/CD9 with phosphatidylserine on the sEV membrane (PS capture ELISA)[38] (Fig. 4d), sandwich ELISA of CD63-CD63 or CD9-CD9, and nluc-based reporter assay (Supplementary Fig. 20). We found that rotenone treatment also reduced the release of CD9⁺ sEV to some extent, but this was also as expected from the GSEAPreranked (Supplementary Fig. 21). These results show that CIBER screening with different sEV markers can identify previously unknown sEV release regulators in a subpopulation-specific manner. On the other hand, the GSEAPreranked suggested that inhibition of proton-transporting V-type ATPase would upregulate the release of CD63⁺ sEVs rather than CD9⁺ sEVs (Fig. 4c, Supplementary Fig. 19, Supplementary Data 6). Indeed, treatment of HEK293T cells with a potent V-type ATPase inhibitor, concanamycin A, dominantly increased CD63⁺ sEV release (Fig. 4e, Supplementary Fig. 20). The result with concanamycin A is in line with the recent observation that CD63⁺ sEVs are "exosomal" (MVB-derived), while CD9⁺ sEVs are more "ectosomal" (plasma membrane-derived) in Hela cells, and V-type ATPase inhibition selectively downregulates degradation of sEVs produced in MVBs[39]. Conversely, we recently showed that inhibition of ATP biosynthesis in mitochondria could promote lysosomal function through mTORC1 and MIT/TFE signaling[40]. So, we believe our findings can be explained as follows: lysosomal function predominantly affects the release of more "exosomal" CD63⁺ sEVs, and thus the inhibition/upregulation of lysosomal function promotes/suppresses the release of CD63⁺ sEVs, respectively (Fig. 4f). This idea is consistent with the results of live-cell imaging showing that CD63 accumulates in intracellular vesicular compartments, while CD9 is mainly localized at the plasma membrane in HEK293T cells used in our study (Supplementary Fig. 22), and the fact that rotenone treatment activates cellular lysosomal function (Supplementary Fig. 23).

As regards other regulators that act differently on the release of CD63⁺ sEVs and CD9⁺ sEVs, the results of GSEAPreranked of CD63/CD9-CIBER in HEK293T cells and tf-idf (term frequency-inverse document frequency) analysis[41] suggested that the downregulation of the cell cycle has a negative effect predominantly on the release of CD9⁺ sEVs (Fig. 5a, Supplementary Fig. 24; note that tf-idf analysis detected a stronger relationship of lysosomal activity with the release of CD63⁺ sEVs again). As expected, halting the cell cycle at the G2/M phase with a cyclin-dependent kinase inhibitor, dinaciclib, suppressed the release of only CD9⁺ sEVs (Fig. 5b, Supplementary Fig. 25). In addition, we synchronized HEK293T cells expressing CD63-nluc or CD9-nluc in the S phase by imposing a double thymidine block (DTB) and periodically estimated the CD63⁺ and CD9⁺ sEVs in the culture supernatant by luminescence measurement while monitoring the progression of the cell cycle after the release from DTB (Fig. 5c, d). We found that the

release of CD9⁺ sEVs, but not CD63⁺ sEVs, was negatively correlated with the ratio of cells at the M phase, and the release of CD9⁺ sEVs seemed to reach the maximum when most of the cells had completed cell division (Fig. 5e). These results are consistent with the idea that the release of CD9⁺ sEVs is synchronized with the cell cycle. Finally, we also found that halting the cell cycle with dinaciclib in two cancer cell lines, SH-SY5Y and HT29 (human colorectal adenocarcinoma-derived), dominantly suppresses the release of CD9⁺ sEVs, suggesting the broad relevance of our biological findings (Fig. 5f). To our knowledge, this study is the first to demonstrate an effect of the cell cycle on sEV heterogeneity.

## Discussion

We have developed a method named CIBER screening that enables the identification of sEV biogenesis/release regulators in a pooled manner and employed it to identify genes and functionally linked gene clusters that affect sEV release. Compared to the conventional one-by-one assay, this pooled assay offers significantly high throughput for the identification of sEV release regulators, with the additional benefit that the effect of sEV re-uptake on the amount of sEVs in the culture media is excluded by randomizing the cellular environment (i.e., in a setting that employs separate wells, downregulation/upregulation of sEV re-uptake might be misinterpreted as upregulated/downregulated sEV release, respectively). An especially noteworthy finding with CIBER screening is that multiple pathways differentially affect the release of CD63⁺ and CD9⁺ sEVs.

To our knowledge, there has been only one previous study on pooled screening of sEV release regulators based on barcoded sEVs, which used miRNAs bearing short EV-targeting nucleotide tags as barcodes loaded in sEVs[9]. However, that study had significant limitations including the necessity for constructing complex custom-made libraries, a high sequence bias for sEV barcoding, the need for a huge amount of cell culture supernatant to harvest sEVs for analysis of the barcodes (>100 L for duplicated screens covering the whole genome), and inability to examine the effect of cell viability and heterogeneity of sEVs. CIBER screening is unique in that CRISPR gRNAs are directly and actively loaded in sEVs as barcodes through interaction with dCas9 fused with an sEV marker. This feature enables efficient and uniform barcoding of sEVs with publicly available gRNA libraries, requiring only ~400 mL of cell culture supernatant (for duplicated screens covering half the genome). In practice, this system allows a single experimenter to implement a large-scale screening within 2 ~ 3 weeks with a ready-to-use gRNA-encoding lentivirus and stable cell lines expressing the necessary components (see Supplementary Fig. 26 for detailed comparison). The sEV-marker-driven gRNA loading into sEVs also allows subpopulation-specific analysis of sEV release for the first time. Furthermore, the ability of CIBER screening to cancel out the changes of barcode abundance in cells upon gene KO is very important, because analysis of gRNA abundance only in the sEV fraction in our screening failed to extract the factors differently affecting CD63⁺ sEV release and CD9⁺ sEV release by GSEAPreranked (Supplementary Fig. 27).

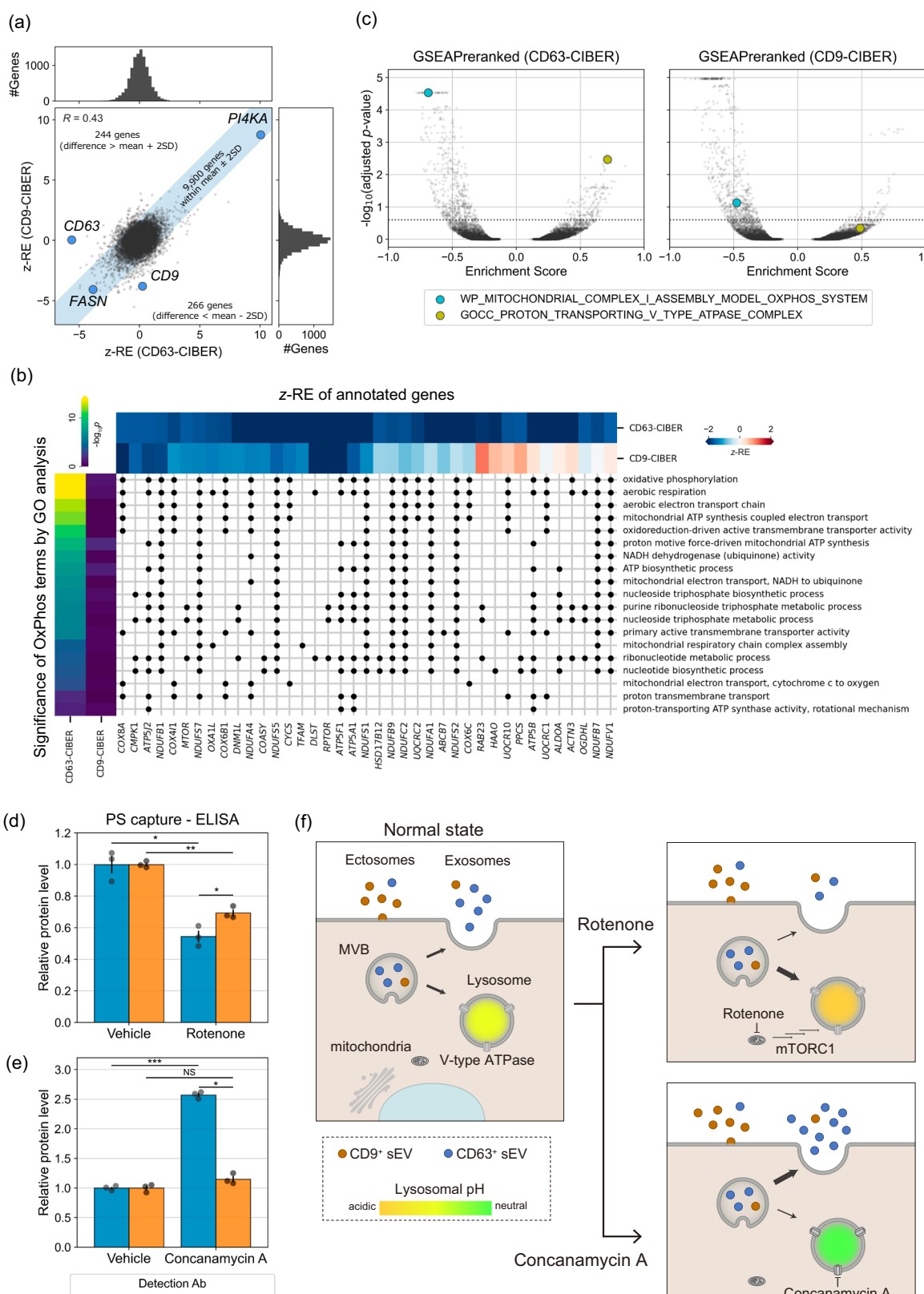

It should be noted that several genes, including Rab family members (e.g. *RAB11*, *RAB27a*, *RAB35*), that reportedly regulate sEV biogenesis[42] were not detected as hits by CIBER screening with the current settings. This could be due to the existence of genes having redundant functions (redundancy of the functions of the Rab proteins has been reported, emphasizing the need for careful consideration of

the outcome of CRISPR screening[43]) or the inefficacy of the gRNAs used. It is also possible that a long time may be required from the time of KO until the effect becomes prominent, or that the KO of a single gene could induce compensation via other pathways, which would also likely be influenced by the assay timeline. Indeed, KO of certain Rab proteins has been reported to gradually kill the cells[43], and we have also

**Fig. 4 | The results of the CD9-CIBER screening and comparison with those of CD63-CIBER screening. a** The pair-wise plot of z-REs obtained from CD63/CD9-CIBER screen for 10,410 genes. The shaded region contains the genes showing the z-RE difference between CD63-CIBER and CD9-CIBER within the mean ±2 SD. **b** Gene-GO term association matrix where black dots are plotted at the intersection if genes are annotated to the terms. The heatmap at the top shows z-RE of each gene annotated to GO terms grouped as OxPhos in Fig. 3i, whose -$\log_{10}$(p-value) (two-tailed Fisher's exact test with Holm correction) are displayed as a heatmap at the left. Only genes with z-REs less than −1.65 in either screening are displayed. **c** Volcano plot of enrichment score of each gene set calculated via GSEAPreranked after ranking all the screened 10,410 genes. The dashed line shows adjusted p-value of 0.25 calculated by two-tailed Kolmogorov–Smirnov test with Benjamini–Hochberg correction. In this data representation, KO or inhibition of components annotated to gene sets in the upper left/upper right area is predicted to downregulate/upregulate the release of each subpopulation of sEVs,

respectively. **d**, **e** Relative abundance of CD63 and CD9 on the surface of sEVs after treatment with rotenone (an inhibitor of mitochondrial complex I) at 10 nM (**d**) or concanamycin A (ConA, a V-type ATPase inhibitor) at 1 nM (**e**) for 24 h. sEVs were captured on Tim4-immobilized wells and detected using either anti-CD63 antibody or anti-CD9 antibody conjugated with horseradish peroxidase. Vehicle: 0.1% DMSO. Error bars represent ±SEM of biological replicates (n = 3). **f** Proposed mechanisms of selective change of the release of CD63⁺ sEVs associated with lysosomal perturbation. Treatment of cells with ConA neutralizes lysosomes, and the contents of MVB tend to evade lysosomal degradation, leading to increased release of "exosomal" CD63⁺ sEVs. Conversely, treatment of cells with rotenone enhances the activity of lysosome, leading to decreased release of exosomal CD63⁺ sEVs. On the other hand, CD9⁺ sEVs are rather "ectosomal", so their release is less affected. p: two-tailed Welch's t-test with Holm correction. *p < 0.05, **p < 0.005, ***p < 0.0005. NS, not significant.

confirmed that the activity of various pathways, especially those predicted to be involved in sEV release processes by CIBER screening, can be influenced by the KO of the relevant genes in a time-dependent manner (Supplementary Fig. 28), suggesting that at least some compensation does occur. From this viewpoint, testing CIBER screening with CRISPR activation or inhibition should be an interesting option to explore other regulators, since the assay timeline with these systems is shorter than that with CRISPR KO screening[44]. At the same time, we wish to emphasize that some false positives and false negatives are inevitable in this kind of large-scale screening. Regarding this point, the application of various exploratory bioinformatic analyses with a mild threshold is a good strategy to avoid missing an important biological pathway, as we have shown in the present work on subpopulation-specific sEV release regulators. Furthermore, it is important to note that it is impossible to predict the hit accuracy of a comprehensive screening platform like CIBER screening before establishing the system. Conducting CIBER screening under a variety of conditions with multiple cell lines would be necessary to address this issue in the future, and this would also be an effective approach for unveiling other previously unknown factors (and gene networks) controlling sEV release.

It may also be necessary to consider the effect of overexpression of dCas9 fused with an EV marker. In the present case, we confirmed that dCas9-fused CD63 and CD9 reached the main destination of their native counterparts and drastically enhanced gRNA loading into sEVs, but nevertheless, a significant portion of the fusion protein remained in the inner membrane, including ER, probably because of slow intracellular trafficking due to the effect of the fusion of a large protein (Supplementary Fig. 29). Though we should emphasize that our major conclusion has been confirmed to be relevant for native sEVs as well, it would be worth optimizing the system in future studies (e.g., finding a small RBP capable of efficiently recruiting gRNA). Precise knock-in (rather than overexpression) of gRNA-recruiting protein would also be an option to secure the optimal expression level for precise intracellular localization of the gRNA-recruiting protein[20].

Despite some issues remain, as discussed above, we believe CIBER screening provides massive information on sEV release that would be inaccessible with conventional methods. Further, the present data should be a useful resource for future studies on sEVs, including more detailed analyses of how each gene regulates sEV release.

For the future, analyses of the detailed actuation mechanism of each hit found by CIBER screening would deepen the understanding of sEV biology. As for the dependency of the release of CD9⁺ sEVs on the cell cycle, CIBER screening suggested that the KO of genes essential for the cell survival generally suppressed CD9⁺ sEV release, so cell division itself (rather than key regulator genes) might influence the release of this subpopulation of sEVs (Supplementary Fig. 30). Also, considering that the growth and malignancy of SH-SY5Y cells[45] and HT29[46] cells are reportedly related to the sEV release process, the different effects of the cell cycle on CD63⁺ and CD9⁺ sEV release observed in these cell

lines raise a question about how cell-cycle-inhibiting anti-cancer drugs might influence the release of a specific subpopulation of sEVs that affects cancer malignancy (note that we also observed a difference between cell types: halting the cell cycle of SH-SY5Y and HT29 cells increased the release of CD63⁺ sEVs (Fig. 5f)). Pursuing this kind of question should deepen our understanding of the contribution of sEV heterogeneity to diseases. Related to the difference between cell types, large-scale parallel CIBER screenings of cell-type-specific regulators of sEV would be an interesting approach to the discovery of drug targets for sEV-related diseases. From this viewpoint, we would like to emphasize that in this work we have already shown the portability of the CIBER screening system to multiple cell lines, as well as the potential of the system to find both sEV regulators that work in multiple cell types and cell-type-specific sEV release regulators. Besides, the knowledge gained by CIBER screening could be applied for the more efficient production of sEV-based next-generation therapeutics with controlled heterogeneity. It might also be possible to apply CIBER screening to other classes of EVs. Indeed, we confirmed that the gRNA barcoding strategy works when dCas9 is fused with other EV proteins, such as ALIX, TSG101 and PTGFRN[47] (Supplementary Fig. 31). Also, the use of protein modifications to efficiently recruit cytosolic protein into sEVs by interaction with the inner leaflet of the sEV membranes would be an interesting approach to assay the release regulators of sEVs focusing on lipid composition rather than focusing on a specific marker protein[48]. Similarly, if we change the dCas9 recruiting protein to a viral protein, it should be possible to assay the regulators of virus production with the same strategy.

From a different viewpoint, the gRNA barcoded sEVs could be potentially used to identify the factors affecting the sEV fate (e.g. cellular uptake, biodistribution, etc) in a high-throughput manner by tracing the fate of the barcodes. Further, the excellent correlation of gRNA compositions in the sEV and cellular fractions in our system implies that this sEV-barcoding system would also be applicable for continuously monitoring cell population dynamics with gRNA composition in sEVs without destroying the cells, allowing for continuous cell-free CRISPR screening (Supplementary Fig. 32; the CIBER screening can be used as a counter assay). Taking into consideration that sEVs protect the enclosed RNA, which reflects the state of the originating cells, from degradation in vivo[49,50], the system could be potentially adapted even for continuous in vivo CRISPR screening. Thus, we envision that CIBER screening will be applicable for a variety of purposes in the future.

## Methods

### Cell culture

HEK293T cells were distributed by RIKEN BRC CELL BANK (RCB2202) (Tsukuba, Japan). SH-SY5Y cells were kind gift from Dr. Yukiko Hori (originally from ATCC (VA, USA, CRL-2266)), HT29 cells were purchased from ATCC (VA, USA, HTB-38). HEK293T and SH-SY5Y were

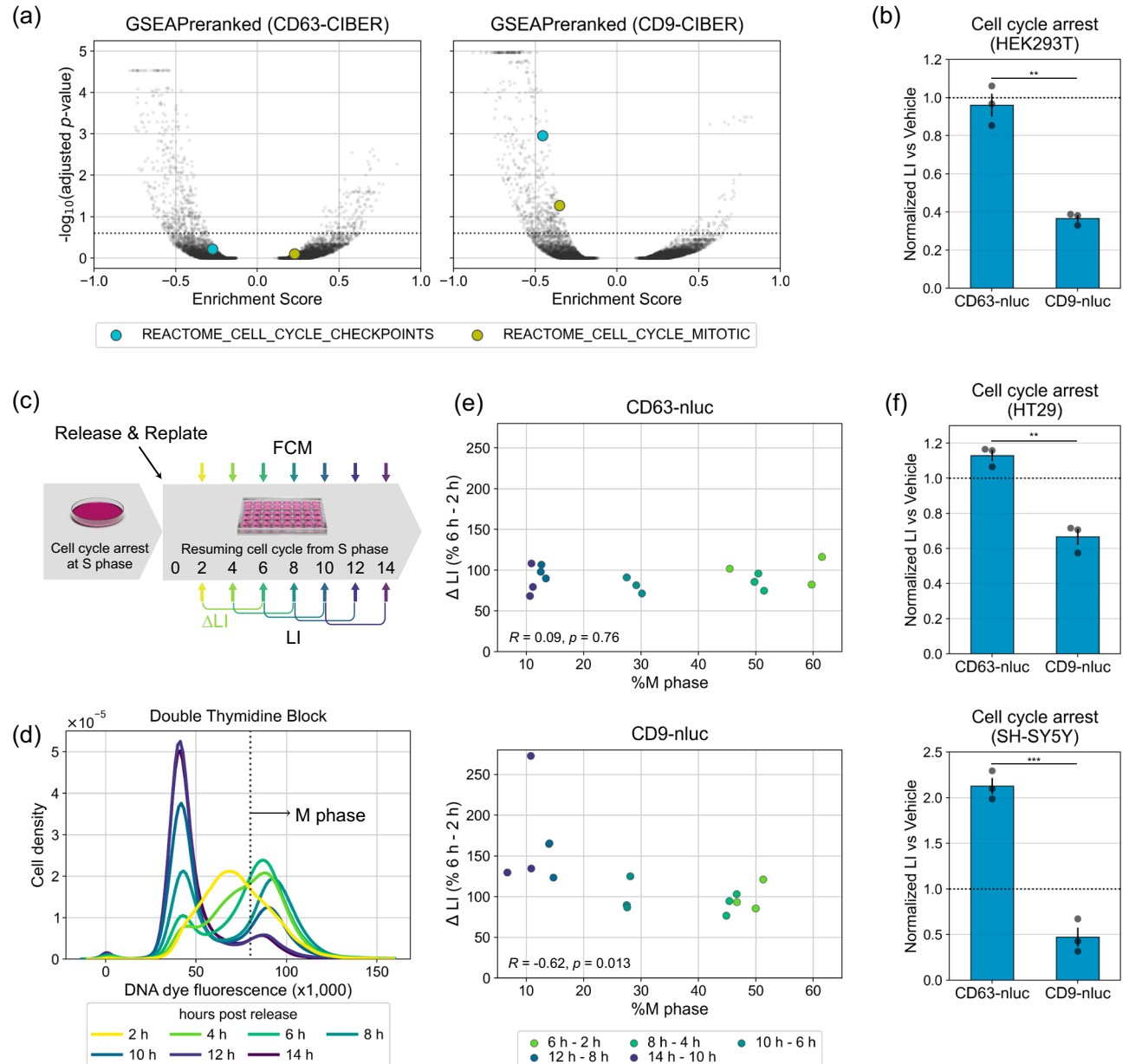

**Fig. 5 | Synchronization of the cell cycle and release of CD9⁺ sEVs. a** Volcano plot of enrichment score calculated via gene set enrichment analysis (GSEA). All the screened 10,410 genes were ranked by $z$-REs and analyzed using GSEAPreranked. Each plot represents a gene set. The dashed line shows adjusted $p$-value of 0.25 calculated by two-tailed Kolmogorov−Smirnov test with Benjamini−Hochberg correction. Detailed results of the GSEA analyses are listed in Supplementary Data 6. **b** Relative release of CD63⁺ or CD9⁺ sEVs during cell cycle arrest in HEK293T cells. HEK293T cells stably expressing CD63-nluc or CD9-nluc were treated with dinaciclib at 10 nM to stop the cell cycle at the G2/M phase for 24 h. Cells were replated and additionally cultured in the presence of dinaciclib for 12 h. LI of CM was measured at 4 and 12 h after replating. Relative sEVs release was calculated as the difference of LI between the two time points. Error bars represent ±SEM of biological replicates ($n = 3$). $p$: two-tailed Welch's $t$-test. **c** Time course of the cell-cycle synchronization assay. HEK293T cells expressing CD63-nluc or CD9-nluc are synchronized at S phase by means of a double thymidine block in bulk culture and then replated into separated wells without thymidine, allowing the cells to resume the cell cycle from the S phase. LI of CM and DNA abundance of the cells were measured at 2, 4, 6, 8, 10, 12, and 14 h after replacing the media. Three wells were used for each time point. **d** Representative result of flow cytometry (FCM) with CD63-nluc expressing cells shown in Fig. 5c. **e** Results of the synchronization assay. The percentage of cells in Fig. 5d with DNA dye fluorescence (×1000) greater than 80 was defined as %M phase. ΔLI is the difference between the LI at a time point and the averaged value 4 hours earlier. $p$: $t$-test for correlation. **f** Results of dinaciclib assays performed with HT29 and SH-SY5Y by the same method as used for Fig. 5b. $p$: two-tailed Welch's $t$-test. *$p < 0.05$, **$p < 0.005$, ***$p < 0.0005$.

maintained in Dulbecco's modified Eagle's medium (DMEM, High Glucose with L-Glutamate, FUJIFILM Wako). HT29 was maintained in McCoy's 5A medium (cat. # 16600082, Gibco). Media were supplemented with 10% (v/v) fetal bovine serum (FBS, Biosera) and 1 % (v/v) penicillin/streptomycin solution (PS, FUJIFILM Wako). Cells were cultured at 37 °C in a humidified atmosphere containing 5% $CO_2$.

## Transfection

Cells were plated at $2.5 \times 10^5$ cells/mL onto a 10 cm dish in 10 mL medium and cultured for 24 hours before transfection. Ten µg of total DNA in 1 mL of plain optiMEM was mixed with 40 µl of Poly-ethylenimine "Max" (PEI, Polyscience #24765, 1 mg/mL in $dH_2O$), briefly vortexed and incubated at room temperature for 15 min. Cell

culture medium was renewed before transfection and DNA/PEI mixture was added dropwise to the culture. After a sufficient cultivation period (typically 8–16 h), the medium was renewed again for downstream applications. Transfections were scaled up or down based on the culture area (cm$^2$) when necessary.

## Establishment and maintenance of stable cell lines

The Sleeping Beauty transposase system[51,52] was mainly used for establishing stably transfected cell lines, except for the lentivirus infection of gRNA cassettes. Cells were transfected with DNA mix containing 50 ng of transposase-encoding plasmid (addgene #34879, pCMV(CAT)T7-SB100) and 950 ng of transposon plasmid on a 12-well plate and selected in antibiotics-containing medium (10 µg/mL blasticidin and 300 µg/mL hygromycin). Sufficiently expanded cells were further sorted by FACS (FACS Aria II or III, BD) and the top 10% of cells with strong fluorescence (encoded in the transposon plasmid and expressed separately from objective protein) were collected. Established cell lines were maintained in the selecting medium (depending on the antibiotic resistance encoded in the transposon plasmid). The correspondence of key proteins and plasmid numbers is as follows: CD63-MS2: pKK47, CD63-dCas9: pKK60, Cas9: pRK300, gRNA #1: pKK90, gRNA #2: pKK209, CD9-dCas9: pKK106, CD63-nluc: pKK108, CD9-nluc: pKK147, CD63-sfGFP: pKK150, CD63-mScarlet: pRK397, CD9-sfGFP: pKK151 (See Supplementary Table 1 for details).

## Plasmid construction

The cloning strategy and oligo DNA used for plasmid construction are described in Supplementary Tables 1 and 2.

## Lentivirus production, titration, and transduction to cells

Production and titration of lentivirus were conducted as previously reported[44] with some modifications. In brief, Lenti-X cells (Clontech) were transfected with pMD2.G (addgene #12259), psPAX2 (addgene #12260) and the pooled gRNA library at the ratio of 1:2:4 (ng) and cultured for 6 hours. Then, the medium was renewed, and culture was continued for 42 hours. Virus-containing medium was passed through a 0.45 µm filter (cat. #SLHVR33RB, Millipore), aliquoted and stored at −80 °C. We performed lentivirus production on a well of a 12-well plate for a single gRNA and on a 500 cm$^2$ dish for DTKP, PROT, ACOC, TMMO libraries[16] (spacer sequences are listed in Supplementary Data 7). For titration, cells mixed with polybrene (cat. #12996-81, nacalai tesque, 8 µg/mL final concentration) were transduced by spinfection (1000 × g for 2 h at 33 °C) with increasing volumes of lentivirus and then selected with 0.3 µg/mL (HEK293T) or 1.5 µg/mL (SH-SY5Y) puromycin for 2–3 days under normal culture conditions. The multiplicity of infection was determined by comparing the viability of selected cells with that of the no-puromycin control using a Cell Counting Kit-8 (cat. #CK04, Dojindo).

For screening, HEK293T cells and SH-SY5Y cells were transduced at the MOI of 0.3 according to the same protocol. The number of cells to be transduced was calculated according to the following equation to ensure that at least 500 cells were transduced with each gRNA in a library: (the number of gRNA in library) ×500/0.3. After the transduction, cells were selected with puromycin for 7 days to achieve maximal knockout and used for sEVs isolation while maintaining 500 cells/gRNA.

## Isolation of sEVs

Cells plated at 1.5–2.0 ×10$^5$ cells/mL were propagated to 70–80% confluency for sEVs production. The medium was changed to optiMEM with 1% (v/v) P/S and cells were cultured for another 24–36 h. The culture supernatant was collected and centrifuged stepwise at 300 × g for 5 min and 2000 × g for 10 min to remove cells and large debris, followed by filtration through a 0.22 µm filter (cat. #SLGV033RS Millipore) to remove small debris. At this time, cells were washed with PBS and lysed in TRIzol (cat. #15596018, Invitrogen) at 10$^7$ cells/mL after collection of

the culture supernatant for screening. Filtered supernatant was transferred to tubes for ultracentrifugation (cat. #344058, Beckman Coulter) to which 200 µL of Optiprep (Abbott Diagnostics Technologies AS) was added at the bottom as a cushion[53]. Small EVs were isolated by ultracentrifugation at 120,000 × g and 4 °C for 120 min using Optima XE-90 equipment with a SW32Ti rotor (Beckman Coulter). Supernatants were discarded using a pipette, leaving 1.5–2.0 mL of the sEVs fraction at the bottom. The pellets were resuspended in vesicle-depleted PBS and centrifuged again with 50 µL of Optiprep cushion. Supernatants were discarded, leaving ~0.5 mL of the sEVs fraction. The resulting pellets were resuspended in vesicle-depleted PBS and transferred into PRO-KEEP Protein Low Binding Tubes (cat. #PK-15C-500N, Watson). We usually resuspended sEVs from 100 mL of culture supernatant in 1 mL of PBS, typically resulting in 1.0 ×10$^{11}$–1.0 ×10$^{12}$ particles/mL. Washed sEVs were kept at 4 °C for short-term storage (up to a week) or flash-frozen and kept at −80 °C for longer-term storage.

## Nanoparticle tracking analysis (NTA)

Nanosight LM10 equipment (Malvern Panalytical) was used for NTA followed by evaluation using the NTA software (ver. 3.2). The analysis was performed using a 488 nm laser with a recording time of 30 s, camera level of 15–16 and detection threshold of 5. Three recordings were sequentially performed for each sample. Samples were diluted to 10$^8$-10$^9$ particles/mL before recording. When measuring culture supernatant, samples were cleared by centrifugation at 300 × g for 5 min, 2000 × g for 10 minutes and 10,000 × g for 30 min before the measurement.

## Evaluation of gRNA amount in sEVs

Wild-type HEK293T cells or HEK293T cells stably expressing CD63-dCas9 were transduced with lentivirus carrying gRNA. After selection, sEVs were isolated from 30 mL of culture supernatant as described above. The concentration of sEVs were measured by NTA. Next, samples were diluted to 1 ×10$^{11}$ particles/mL. Three µL aliquots of diluted samples were lysed by mixing with 3 µL of 0.1% Triton X-100 in Nuclease-Free Water (NFW, cat. #AM9937, Invitrogen) and incubating for 10 min at room temperature. The Ct values of samples were determined by quantitative PCR (qPCR) using a Luna Universal Probe One-Step RT-qPCR Kit (cat. #E3006, New England Biolabs) with LightCycler 480 (Roche Diagnostics) according to the manufacturers' instructions. The sequences of primer and probe (IDT PrimeTime Mini qPCR assay) are listed on Supplementary Table 2. The same reaction was performed with each gRNA-containing plasmid (in pCRISPRia-v2) at 0.0001–1000 pg/µL to obtain a standard plot which was used to calculate the absolute number of gRNA molecules in each sample. The number of gRNA molecules was divided by the number of particles in the PCR reaction to calculate copies/particle (Fig. 1c).

## Sample preparation for NGS

**-RNA extraction and DNA digestion.** For cellular RNA, total RNA was extracted from at least 500 cells/gRNA according to the manufacturer's protocol and dissolved in 40 µL of NFW. Contaminant DNA in 50 µg of extracted RNA was digested with DNaseI (cat. #2270A, Takara) according to the manufacturer's protocol. After DNA digestion, the reaction was quenched by adding 5 µL of EDTA-2Na [250 mM] followed by incubation at 80 °C for 2 min. Then 45 µL of NFW, 10 µL of NaOAc [3 M] and 250 µL of chilled EtOH were added to precipitate nucleic acids, and the mixture was incubated at −80 °C for 20 min, and centrifuged at 120,000 × g and 4 °C for 30 min. The resulting pellet was washed with 750 µL of chilled 70% EtOH and resuspended in 40 µL of NFW. Two µg of RNA was used for reverse transcription (RT).

For sEVs, total RNA was extracted from 250 µL of sEVs solution with 1 mL of TRIzol. The RNA-containing phase was mixed with GlycoBlue™ Coprecipitant (cat. #AM9515, Invitrogen) before isopropanol precipitation. The RNA pellet was dissolved in 5 µl of NFW and used for RT.

**-RT of transcribed gRNA (SMART technology[54]) and clean-up.** Five µL of RNA was mixed with 0.5 µL of oKK145 [12 µM], incubated at 70 °C for 3 min, put on ice to form RNA/primer complex and mixed with 4.5 µL of RT solution {0.5 µL of SMARTScribe RTase (cat. #639536, Takara), 2 µL of 5x first strand buffer (supplied with RTase), 0.25 µL of DTT (supplied with RTase), 1 µL of dNTPs (cat. #N0447, NEB), 0.5 µL of custom LNA-TSO [12 µM] (cat. # 339412, Qiagen) and 0.25 µl of RNase inhibitor (cat. #2313A, Takara)}. The RT mixture was incubated at 42 °C for 120 min and heated at 70 °C for 10 min. After the RT reaction, 18 µL of AMPure XP (cat. #BC-A63881, Beckman Coulter) was added to the solution and incubated for 10 min at room temperature. The cDNA bound to beads was separated on magnet rack and subjected to tagging PCR without washing. For MS2-gRNA, Oligo #4 was used instead of oKK145.

**-Tagging PCR.** To the bead-bound DNA, 25 µL of PCR solution {0.15 µL of oKK147 [50 µM], 0.15 µL of (each of Oligo #6-Oligo #17) [50 µM], 12.5 µL of Tks Gflex DNA polymerase Low DNA (cat. #R091A, Takara) and 12.2 µL of NFW} was added. PCR conditions were as follows: 95 °C for 1 min, 20 cycles of (98 °C for 10 s, 65 °C for 30 s, 68 °C for 30 s), 68 °C for 30 seconds and 4 °C hold. The PCR product (NGS sample) was purified with 35 µL of AMPure XP and eluted in 20 µL NFW. The concentration of NGS sample was measured by qPCR using a primer set of oKK120/oKK121. The standard curve was obtained using a commercial control (*E. coli* DH10B library control in Ion Library Taq-Man™ Quantitation Kit, Thermo Fisher Scientific Inc.) diluted to 6.8, 0.68, 0.068, 0.0068, and 0.00068 pM. For MS2-gRNA, Oligo #5 was used instead of oKK147.

**Electrophoresis on the Agilent 2100 Bioanalyzer.** The NGS sample was diluted with TE buffer and assayed using a High Sensitivity DNA Kit (cat. #5067-4626, Agilent Technologies) on a Bioanalyzer 2100. Raw data was exported as a csv file using the 2100 expert software.

**NGS using Ion Proton.** Each sample was diluted to 50 pM and sequenced by an Ion Proton instrument (Thermo Fisher Scientific Inc.) using an Ion PI Hi-Q Chef Kit (cat. #A27198) and Ion PI Chip Kit v3 (cat. #A26771). One chip generally offers $10^8$ valid reads, so we pooled 4 samples (25,000–30,000 gRNAs/sample) aiming for >500 reads/ gRNA for each sequencing. The chip was prepared using an Ion Chef instrument (Thermo Fisher Scientific Inc.). FileExporter was used to export fastq files for downstream analysis.

**NGS data processing.** We share a Python script count_barcodes.py (a modified version of count_spacers.py[44]) for barcode counting (see the section of Code availability). We prepared a.xlsx file containing the barcode id in each line to be referenced with a header named 'id'. The format of the barcode id should be '{target gene}_{spacer sequence}_{additional information}' (e.g., 'AADACL2_GAAAGTCAGAAACCCGA_2832.7_DTKP'). Each read is assigned to a corresponding barcode if the 8 bp of scaffold sequence flanked to the spacer (GTTTAAGA; substituted to KEY in the script) is detected and 17 bp of the sequence upstream of KEY is identical to the spacer sequence. All the generated count files are available in Supplementary Data 2. Relevant statistics are also available in Supplementary Data 1.

**Calculation of z-normalized release effect (z-RE).** We share a Python script calculate_zRE.py for calculation of z-normalized release effect (see the section of Code availability). Raw read counts are normalized to calculate nRC so that average nRC in a sample is 1. Each barcode would have 4 nRCs depending on the sample origin; (cell/sEVs) × (Cas9$^+$/Cas9$^-$). Barcodes with nRC lower than 0.05 in at least one sample were excluded from downstream calculation and after this exclusion, any genes with less than 3 barcodes were also excluded. $FC_{cells}$ and $FC_{sEVs}$ are calculated for each barcode as $\log_2(nRC_{cell \times Cas9+}/$

$nRC_{cell \times Cas9-})$ and $\log_2(nRC_{sEVs \times Cas9+}/nRC_{sEVs \times Cas9-})$, respectively. Linear regression was performed on the scatter plot with $FC_{cells}$ on the *x*-axis and $FC_{sEVs}$ on *y*-axis. The release effects (RE) for each barcode were tentatively calculated as residues from the regression line (RE at gRNA level). Among every barcode group targeting the same gene, the barcodes with highest/lowest RE were excluded from the scatter plot and the regression line was drawn again. RE at the gRNA level was calculated again using the new regression line. RE at the gene level is the median value of REs of barcodes targeting the gene. REs at the gene level were *z*-normalized for each subpool library to calculate *z*-REs. Genes with *z*-REs larger than 1.65 are regarded as upper hits and genes with *z*-REs lower than −1.65 are regarded as lower hits in this paper.

**Inhibition assay with GSK-A1 and C75.** Cells were plated onto 48-well plates ($4.5 \times 10^4$ cells in 300 µL of supplemented media) and allowed to expand to around 70% confluency. Then, media were replaced with optiMEM (1% PS) containing GSK-A1 (cat. #SYN-1219-M001, AdipoGen) or C75 (cat. #10005270, Cayman Chemical Co.). Cells were treated for 24 h for GSK-A1 or for 30 min followed by transfer to supplemented media without inhibitor for C75. Wild-type HEK293T cells were used for nanoparticle tracking analysis and HEK293T cells and SH-SY5Y cells expressing CD63-nluc or CD9-nluc were used for nluc-based reporter assay.

For reporter assay, culture supernatant was centrifuged at $300 \times g$ for 5 min and $2000 \times g$ for 10 min to remove cell debris. The resulting supernatant was diluted at 1:50 with PBS and used for luminescence measurement with the Nano-Glo Luciferase Assay System (cat. #N1110, Promega) according to the manufacturer's protocol. The protein concentration of the cellular fraction was used to normalize the measured luminescence intensity (LI). After the removal of the culture supernatant, 75 µL of CelLytic M (cat. #C2978, Sigma Aldrich) was added to each well. The plate was gently agitated for 20 min at room temperature and the protein concentration was measured using a Pierce BCA Protein Assay Kit (cat. #0023227, Invitrogen) according to the manufacturer's protocol. The LI was divided by the protein concentration of corresponding well to calculate normalized LI.

**Hit validation with siRNA.** A reverse transfection protocol was adopted for siRNA transfection. The transfection mix was prepared by mixing 2 µL of Lipofectamine RNAiMAX Transfection Reagent (cat. #13778030, Thermo Fisher Scientific) in 98 µL of plain optiMEM and 20 µL of siRNA [500 nM] in 80 µL of plain optiMEM. Two hundred µL of transfection mix was incubated for 5 min at room temperature, and mixed with 800 µL of cell suspension at $2.0 \times 10^5$ cells/mL and the cells were plated onto a well of a 12-well plate. Media were refreshed after 24 h. Culture was continued for another 24 hours, and the cells were passaged to 3 wells of a 48-well plate and cultured for 24 h. Media were replaced with optiMEM (1% PS) and incubation was continued for 24 h. After the final incubation, the culture supernatant was harvested and processed as described above. The 27-mer synthetic double-strand RNAs (DsiRNA, manufactured by IDT) were used for knocking down OSBP, TMED10 and GOLGA2. A pool of siRNA (siPOOL, manufactured by siTOOLS BIOTECH) was used for NDUFS1. Conventional 21-mer siRNAs (manufactured by Bioneer) were used for other genes. The source information of siRNA is listed in Supplementary Table 4. All the siRNAs were used as mixture of siRNA #1-siRNA #3.

**Protein-protein interaction.** Upper hits and lower hits were separately queried in the STRING database via the StringApp plugin (https://apps. cytoscape.org/apps/stringapp) on Cytoscape[55] (ver. 3.9.1, https:// cytoscape.org/). Interaction maps were drawn with default settings. Gene products were represented as nodes and connected with edges to each other if STRING analysis predicted interactions. Nodes without any interactions were not displayed. Node filling colors were changed

to show *z*-REs. The borders of the nodes for genes validated in Fig. 3f (*OSBP, FASN, TMED10, GOLGA2, NDUFS1, PTPN23, KIAA1109, CAB39, VPS28, PI4KA*) were changed to thick red. The borders of the nodes directly connected to validated genes were changed to thick gray.

**Gene ontology (GO) analysis.** Enrichment analysis against the GO Biological Process data set was performed using the ClueGO plugin (ver. 2.5.9, https://apps.cytoscape.org/apps/cluego) on Cytoscape (ver. 3.9.1). Analysis parameters were as follows; Marker Lists: upper hits or lower hits, Ontology:

GO_BiologicalProcess-EBI-UniProt-GOA-ACAP-ARAP_25.05.2022_00h00, *p*-value cutoff = 0.05, Correction Method = Bonferroni step down, Min GO Level = 6, Max GO Level = 13, Number of Genes = 2, Min Percentage = 5.0, GO Fusion = true, GO Group = true, Kappa Score Threshold = 0.4. We used raw output file (Supplementary Data 5, sheet 'CD63low', 'CD63up', 'CD9low' and 'CD9up') to draw Fig. 3i and Supplementary Fig. 16b. For Fig. 4b, OxPhos terms were individually analyzed with CD9-CIBER hits as query. The output results were added to the raw output file on sheet 'CD9low' and *p*-value adjustment by Bonferroni step down was manually performed to calculate adjusted *p*-value.

**Hit validation with PS (phosphatidylserine) capture ELISA.** HEK293T cells were plated onto 48-well plates ($4.5 \times 10^4$ cells in 300 μL of supplemented media). Cells were allowed to expand to around 70% confluency. Then, media were replaced with 500 μL of supplemented media containing rotenone (cat. #R0090, Tokyo Chemical Industry) at 10 nM or concanamycin A (cat. #BVT-0237-C025, AdipoGen) at 1 nM. Cells were treated for 24 hours, then the culture media were harvested and cleared by stepwise centrifugation of $300 \times g$ for 5 min, $2000 \times g$ for 10 min and $10,000 \times g$ for 30 min at 4 °C.

The expression levels of CD63 and CD9 on the sEVs surface were measured using a PS Capture Exosome ELISA Kit (cat. #298-80601, Fujifilm) according to the manufacturer's protocol. Samples were diluted 1:10 with Reaction Buffer. Anti-CD63 antibody was supplied with the kit. Biotinylated anti-CD9 antibody was purchased from Fujifilm (cat. #013-27951, lot CAE1209, Fujifilm) and used at 240 ng/mL diluted with Reaction Buffer.

**GSEAPraranked analysis.** GSEAPreranked was performed with GSEA software (ver. 4.2.3, https://www.gsea-msigdb.org/gsea/index.jsp) using the following .gmt files (gene sets); h.all.v2023.1.Hs.symbols.gmt [Hallmarks], c2.all.v2023.1.Hs.symbols.gmt [Curated], c5.all.v2023.1.Hs.symbols.gmt [Gene ontology]. Gene names and corresponding *z*-REs were queried to run GSEAPreranked with the default setting of 1000 permutations and No_Collapse.

**Cell cycle arrest with dinaciclib.** Cells expressing CD63-nluc or CD9-nluc were plated onto a 10 cm dish (HEK293T) or 48-well plate (HT29 and SH-SY5Y) and allowed to expand to around 70% confluency, then treated with dinaciclib (cat. # D479725, Toronto Research Chemicals) at 10 nM or vehicle (0.1% DMSO) in supplemented media for 24 h; since HEK293T cells are easily detached from culture surface after media replacement and the numbers of remaining cells are difficult to control, we treated HEK293T on a dish and replated them to a 48-well plate after cell counting. Media were refreshed to the same-conditioned one and cells were incubated for another 4 or 12 h. After each additional incubation, the culture supernatant was harvested and used for LI measurement as described above. Relative EV release was calculated by subtracting the LI at 4 h from the LI at 12 h and normalizing the value to the mean value under vehicle conditions.

Immediately after media collection, cells were washed with 100 μL of PBS, resuspended in 250 μL of PBS, mixed with 2.5 μL of Cell Cycle Assay Solution Blue (cat. #C549, Dojinbo) and incubated for 15 min at 37 °C in the dark to stain nuclei. Stained cells were directly subjected to

flow cytometry (FCM) analysis on a BD LSR II (Becton, Dickinson and Company) to measure the DNA amount in individual cells. FCM data was processed on FACS Diva software (ver. 4.1).

**Double thymidine block.** HEK293T cells were synchronized at the G1/S boundary by the double thymidine block (DTB) method as previously described[56]. DTB was performed by adding thymidine to a final concentration of 2 mM to a culture of $2.0 \times 10^6$ HEK293T cells expressing CD63-nluc or CD9-nluc plated onto a 10 cm dish 24 h prior to thymidine addition, incubating each dish for 18 h, refreshing the media after washing the dish 3 times with PBS and incubating for 9 h, and treating cells again with thymidine at 2 mM for 18 hours to complete cellular synchronization.

After the DTB, cells were detached from the dish, resuspended in supplemented media without thymidine, diluted to $1.0 \times 10^6$ cells/mL, and replated (500 μL) onto 21 wells of a 48-well plate. At 2, 4, 6, 8, 10, 12 and 14 hours after replating, the LI of culture media and DNA amount in cells were measured in the same way as described for dinaciclib assay (*n* = 3). ΔLI is the relative sEV amount released during the 4-h interval, calculated by subtracting the mean LI at 2, 4, 6, 8, 10 h from LI of every replicate at 6, 8, 10, 12, 14 h, respectively, and normalizing all values to mean ΔLI between 6 h and 2 h. FCM data processed on FACS Diva software were further analyzed using a Python package FlowCal[57] (https://taborlab.github.io/FlowCal/) to calculate the ratio of cells at M phase. The G1 peak appeared around 45,000 < DNA dye fluorescence <50,000 and the G2/M peak appeared around 85,000 < DNA dye fluorescence <90,000, so we assigned cells with DNA dye fluorescence larger than 80,000 to M phase.

**Western blotting.** Whole cell lysates (WCL) were prepared by lysing cells cultured on dishes with CelLytic M (cat. #C2978, Sigma-Aldrich). For FACS-collected cells, $1.0 \times 10^6$ cells were lysed with 100 μL of CelLytic M. Ten μg of WCL or 0.5 μg of sEV (2-3 $\times 10^9$ particles, isolated by ultracentrifugation from culture supernatants) were mixed with 4× Laemmli Sample Buffer (Cat. #1610747. BIORAD) without (for CD63 and CD9 detection) or with (for the others) β-mercaptoethanol. Samples were denatured under mild (37 °C for 15 min, for CD63 and CD9 detection) or standard (95 °C for 5 min, for the others) conditions. Proteins were separated by sodium dodecyl sulfate – poly-acrylamide gel electrophoresis (SDS-PAGE) and transferred to polyvinylidene difluoride (PVDF) membrane (cat. #34002, Invitrogen). Membranes were blocked with Blocking One (cat. #03953, nacalai tesque) and probed with primary antibody diluted with 10% Blocking One/TBS-T followed by incubation with HRP-conjugated secondary antibody (cat. #7074, lot 33 or #7076, lot 38, Cell Signaling Technology) diluted at 1:2000 with 10% Blocking One/TBS-T. The dilution factors of primary antibodies are as follows: anti-CD63 antibody (1:1000, cat. #SHI-EXO-M02, lot 23H23CB, CosmoBio), anti-CD9 antibody (1:1000, cat. #SHI-EXO-M01, lot 28G23CB, CosmoBio), anti-dCas9 antibody (1:1000, cat. #A-9000-010, lot 2203061, Epigen Tek), anti-calnexin antibody (1:1000, cat. #EPR3633(2), lot GR3416744-20, abcam), anti-RAB27A antibody (1:500, cat. #95394, lot 1, Cell Signaling Technology), anti-ALIX antibody (1:1000, cat. #12422-1-AP, lot 00115246, Proteintech), anti-β-actin antibody (1:1000, cat. #4970, lot 19, Cell Signaling Technology). Proteins were detected using enhanced chemiluminescence (ECL) substrate (cat. #1705060, BIORAD) and iBright FL1500 Imaging Systems (Thermo Fisher Scientific). Antibodies were removed from membrane using stripping buffer (cat. #T7135A, Takara) for subsequent probing with another antibody.

**Transmission electron microscopy.** sEVs were isolated from culture supernatant as described above. Three μL of sEV suspension at $1 \times 10^{11}$ particles/mL was placed on a Formvar/Carbon coated copper mesh-grid and allowed to adsorb for 90 s. Samples were contrasted by 3 cycles of dipping grids into a drop of 1% uranyl acetate and removing excess liquid with a filter paper. Grids were examined at 100 kV using a

transmission electron microscope JEM-1400 (JEOL) equipped with an EM-14661 camera.

**Creation of the MS2-gRNA library.** The MS2-gRNA library was created by replacing the scaffold region of the hCRISPRa-v2 h6 library (addgene #83985, 13,145 gRNAs) with the MS2-gRNA scaffold in pKK49. Three μg of hCRISPRa-v2 h6 library and 5 μg of pKK49 were digested with BlpI/NheI at 37 °C for 3 h. After agarose gel purification, each fragment was eluted with 24 μL of MilliQ. Five μL of spacer-containing fragments and 8 μL of scaffold-containing fragments were ligated with 13 μL of Ligation high Ver.2 (cat. # LGK-201, TOYOBO) at 16 °C overnight. After EtOH precipitation, the ligated product was amplified following a protocol provided by Joung et al.[44].

The diversity of gRNA in the created library was quantified by NGS. The spacer regions were amplified by PCR in a 50 μL reaction mixture composed of 45 μL of Platinum PCR SuperMix High Fidelity (cat. #12532016, Invitrogen), 4 μL of the library [35 ng/μL], 0.5 μL of Oligo #5 [5 μM] and 0.5 μL of Oligo #18 [5 μM]. PCR conditions were as follows: 94 °C for 2 min, 40 cycles of (94 °C for 30 s, 65 °C for 30 s, 68 °C for 30 s), 68 °C for 30 s and 4 °C hold. The PCR product was purified using AMPure XP according to the manufacturer's protocol and eluted in 20 μL of nuclease-free water. Sequencing and downstream analysis were performed as described above.

**Flow cytometry analysis of knockout efficiency.** The gRNA-transferring virus were prepared with gCtrl (pKK90) or gBFP (pKK209) on a well of a 12-well plate as described above. Wild-type HEK293T cells, HEK293T cells expressing either of Cas9 or CD63-dCas9 and HEK293T cells expressing both Cas9 and CD63-dCas9 were transduced with the virus and selected on puromycin at 0.3 μg/mL for 1 week. Cells were subjected to flow cytometry (FCM) analysis on a BD LSR II (Becton, Dickinson and Company) to measure the BFP intensity using the Pacific Blue channel. FCM data was processed using FACS Diva software (ver. 4.1).

**Quantitative polymerase chain reaction (qPCR).** Transfection was performed on a 96-well plate at 1/16 scale of the 12-well plate format described in the methods related to Fig. 3f. Cells were cultured for 24 h and then lysed using a SuperPrep II Cell Lysis Kit for qPCR (cat. #SCQ-501, Toyobo) according to manufacturer's protocol. Two μL of lysate was applied for 1 step qPCR using RNA-direct SYBR Green Realtime PCR Master Mix (cat. #QRT-201, Toyobo. Twenty μL of reaction mixture consisted of 2 μL of lysate, 10 μL of Master Mix, 1 μL of Mn(OAc)$_2$ at 50 mM, 0.1 μL of forward primer at 50 μM, 0.1 μL of reverse primer at 50 μM and NFW up to 20 μL). PCR conditions were as follows; 90 °C for 30 s, 61 °C for 20 min, 95 °C for 30 s, 45 cycles of (95 °C for 5 s, 55 °C for 10 s and 74 °C for 15 s). Relative mRNA level was calculated using the $2^{-\Delta\Delta Ct}$ method with GAPDH as an internal control. Primer sequences are listed in Supplementary Table 2.

**Hit validation with sandwich ELISA.** Cells were treated according to the same protocol as described in the Method section of PS capture ELISA. The expression levels of CD63 and CD9 on the sEVs surface were measured using a CD63-Capture Human Exosome ELISA Kit (cat. #290-83601, Fujifilm) or CD9-Capture Human Exosome ELISA Kit (cat. #296-83701, Fujifilm) according to the manufacturer's protocol.

**Localization study of CD63-mScarlet and CD9-sfGFP.** 40,000 HEK293T cells in 200 μL of culture medium were reverse-transfected with 100 ng of pKK151 (CD9-sfGFP) and 100 ng of pRK397 (CD63-mScarlet) on an 8-well chamber (cat. #ib80826, Ibidi). After 24 h of incubation, cells were visualized by a confocal fluorescence microscope (TCS SP8, Leica). The excitation and emission wavelengths can be found in the figure legends.

**Visualization of cellular lysosomal activity.** Wells of an 8-well chamber were coated with 200 μL of collagen solution (cat. #TMTCC-050, Toyobo) for 1.5 h at room temperature and washed with 200 μL of PBS twice. HEK293T cells were plated (3.0 ×10⁴ cells in 200 μL of supplemented media) on the coated wells and allowed to expand to around 70% confluency. Media were changed to 200 μL of optiMEM containing either of vehicle (0.1% DMSO), rotenone at 10 nM or concanamycin A at 1 nM. After 24 h of incubation, cells were stained with Hoechst33342 (Invitrogen, H1399) at 10 μg/mL for 10 min in DPBS and then LysoTracker™ Red DND-99 (cat. #L7528, Invitrogen) at 50 nM for 30 min in supplemented media before visualization by a confocal fluorescence microscope. The excitation and emission wavelengths can be found in the figure legends.

**Immunofluorescence imaging.** Five thousand cells were plated on an 18-well chamber (cat. #ib81816, Ibidi) coated with collagen, as described above. After two days of incubation, cells were washed with PBS, fixed with 4% PFA at room temperature for 10 min, washed 3 times with PBS, permeabilized with 0.1% Triton X-100/PBS at room temperature for 10 min and blocked with Blocking One at room temperature for 30 min. The permeabilization step was skipped for Supplementary Fig. 29c. After the removal of blocking solution, cells were probed with primary antibody diluted with 10% Blocking One/PBS at room temperature for an hour or 4 °C overnight. The dilution factors of primary antibodies are as follows: anti-CD63 antibody (1:100, cat. #SHI-EXO-M02, Lot 23H23CB, CosmoBio), anti-CD9 antibody (1:100, cat. #SHI-EXO-M01, lot 28G23CB, CosmoBio), anti-dCas9 antibody (1:100, cat. #A-9000-010, lot 2203061, Epigen Tek), anti-GFP antibody (1:500, cat. #598, MBL). Cells were washed 3 times with PBS and then stained with secondary antibody (cat. #A32740, lot YI378038 or #A32723, lot YJ383140 Invitrogen) diluted at 1:400 with 10% Blocking One/PBS along with 0.3 μM of DAPI at room temperature for an hour, protected from light. Cells were visualized by a confocal microscope, Leica TCS SP8. The excitation and emission wavelengths can be found in the figure legends.

**RNA sequencing.** Six million HEK293T cells expressing Cas9 were transduced with lentivirus for the expression of gRNA at MOI = 0.3 ($n = 1$). After 3, 5 or 7 days of selection under puromycin, 1.0 ×10⁶ of BFP⁺ cells were collected by FACS. All the collected cells were washed once with PBS, pelleted and lysed with 400 uL of TRIzol for RNA extraction. Contaminant DNA was digested as described in Method section. RNA samples were divided here for three technical replicates. Messenger RNA is enriched from resulting RNA using NEBNext Poly(A) mRNA Magnetic Isolation Module (cat. #E7490, NEB). The RNA fragmentation step was incubation at 94 °C for 15 minutes. NEBNext Ultra II Directional RNA Library Prep Kit (cat. #E7760, NEB) and NEBNext® Multiplex Oligos for Illumina® (96 Unique Dual Index Primer Pairs Set 2) (cat. #E6442, NEB) were used for library preparation. The number of PCR cycles was 9. All the procedures were performed according to the manufacturer's protocol at the scale of half with one modification of using AMPure XP in place of SPRIselect. Quality control was performed on the Agilent 2100 Bioanalyzer system as described in the Method section. Samples were diluted to 2 nM based on the concentration determined by the Bioanalyzer and pooled. NextSeq 2000 P3 Reagent Cartridge (50 cycles, cat. #20046810, illumina) was used for single-end read on the NextSeq 2000 platform. Reads were aligned to UCSC GRCh38/hg38 using STAR (ver. 2.7.10) and featureCounts (ver. 2.0.1). The relevant data is available DDBJ database under accession code PRJDB17057.

**TF-IDF analysis.** Terms with adjusted $p$-value lower than 0.25 in the results of GSEAPreranked (Supplementary Data 6) were collected and concatenated to prepare a document-like array. All the underscores in the array were replaced by spaces and the first words of each term showing ontology were removed. The TF-IDF scores for each word were calculated with the following equation: TF-IDF$(t, d) = $ TF$(t, d) \times ($log$_{10}(2/$

DF($t$)) + 1), where TF($t$, $d$) indicates the number of the word $t$ in a document $d$ and DF($t$) indicates the number of documents that contains the word $t$. Any 0 scores are replaced by (minimal TF-IDF score in a document)/2 to avoid division by zero in calculating $\log_2$(TF-IDF ratio).

**Analysis for the application to cell-free CRISPR screening**

Raw read counts were analyzed to calculate z-LFC using MAGeCK software[58] (ver 0.5.9.4, https://sourceforge.net/projects/mageck/files/0.5/mageck-0.5.9.4.tar.gz/download). The cellular counts from Cas9$^+$ samples were used as the treated condition and those from Cas9$^-$ samples were used as the control condition to calculate LFC$_{cells}$. The sEVs counts were also processed similarly to calculate LFC$_{sEVs}$. The LFCs for each subpool library were $z$-normalized before being combined.

**Statistics & reproducibility**

Unless otherwise stated, all computational and statistical analyses in this study were performed using Excel, Python or R. Statistical details for each experiment can be found in the figure legends or corresponding method section and Source Data files. Differences with a (adjusted) $p$-value < 0.05 were considered significant. We included all the data for the analysis (with some exception for analyzing the NGS data. See Methods for the detail). We confirmed that all the data is reproducible. No statistical method was used to predetermine sample size. Sample size was generally chosen based on our experience and the standard practices in the field (typically 3 or more biological replicates for biochemical assays).

**Reporting summary**

Further information on research design is available in the Nature Portfolio Reporting Summary linked to this article.

## Data availability

All data generated in this study is available in the Source Data, Supplementary Data, and public repositories (DDBJ database under accession code PRJDB17057). Sequence data of relevant plasmids have been deposited in GenBank: pKK60 (PQ146490), pRK300 (PQ146491), pKK106 (PQ146492), pKK108 (PQ146493), pKK147 (PQ146494), pKK272 (PQ146495]), pKK273 (PQ146496), pKK274 (PQ146497). Source data are provided with this paper.

## Code availability

We share python scripts used in this study via GitHub (https://github.com/Ryosuke-Kojima/CIBER-screening-paper) and Zenodo (https://zenodo.org/records/13906513).

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

## Acknowledgements

We thank Prof. Kenzo Hirose, Dr. Shigeyuki Namiki and Dr. Daisuke Asanuma for providing the infrastructure necessary to conduct experiments using lentivirus, Dr. Kiyoshi Yamaguchi for the help for using Ion Proton, Dr. Hiroki Sugishita for help in conducting RNA-seq experiments, Ms. Toshie Furuya, Dr. Chieko Saito, and Dr. Masahide Kikkawa for help with TEM analysis, Dr. Reiko Tsuchiya for help in conducting immunofluorescence and western blot experiments, and the suppliers of the Addgene constructs used in this study. This work was supported by the Japan Science and Technology Agency (JST) PRESTO program (JPMJPR17H5 to R.K.), JST FOREST program (JPMJFR214N to R.K.), ATI research grant (to R.K.), HFSP Career Development Award (CDA-00008/2019-C to R.K.), JST CREST program (JPMJCR19H1 to R.K and S.O., JPMJCR23B7 to R.K.), and JSPS KAKENHI (Grant-in-Aid for Transformative Research Areas, 24H00868 to R.K.). TEM analysis was supported by Research Support Project for Life Science and Drug Discovery (Basis for Supporting Innovative Drug Discovery and Life Science Research (BINDS)) from AMED under Grant Number JP23ama121002. K.K. was supported by a Grant-in-Aid for JSPS fellows and WINGS-LST program of the University of Tokyo.

## Author contributions

K.K. and R.K. designed and conducted the majority of the experiments. K.K., R.K., and T.M. performed analysis and visualization of the screening data with a custom-made analysis pipeline. K.H., C.O., M.K., S.O., and Y.U. advised on the experiments. K.K. and R.K. co-wrote the manuscript with input from all the authors. R.K. conceived, directed, and supervised the entire project.

## Competing interests

K.K., Y.U., R.K. have filed a patent for the screening system of sEV release regulators (WO2021095842 (published), application by the University of Tokyo). The other authors report no competing interests.
