## [Transparent Peer Review file · Nature Communications]

Barcoding of small extracellular vesicles with CRISPR-gRNA enables comprehensive, subpopulation-specific analysis of their biogenesis/release regulators

Corresponding Author: Professor Ryosuke Kojima

Version 0:

Reviewer comments:

Reviewer #1

(Remarks to the Author)

The manuscript by Kunitake et al. describes a Cas9/gRNA-based barcoding system that allows a pooled screening approach to identify genes involved in EV release. Biogenesis of EVs remains incompletely understood, and this work offers an elegant, well-designed system to address this. The manuscript is generally well written, and its conclusions sufficiently supported by data. However, despite identification of novel sEV release regulators, the work remains largely descriptive, and identification of novel molecular mechanisms underlying EV release is lacking, which may somewhat limit the impact of this work. Furthermore, it contains several unclarities that need to be addressed.

Specific comments:

- EV isolation and characterization. The isolation procedure methods lack detail (e.g. g-forces used, UC tube types, rotor info). EV characterization is insufficient. The authors should, according to MISEV guidelines, at least characterize EV and potential contaminant marker expression compared to cell lysate expression (e.g. using Western Blot) and perform some type of EV imaging (e.g. EM).
- As discussed by the authors, many of the proteins previously identified to play roles in EV release (Rab27a, Alix, nSmase2, etc) were not picked up in the screen. This requires more attention and discussion. What effects of knockout of these proteins on EV release were found in the screen? What information has been shown by others for the cell line used in this work (HEK293T cells)? And what differences (e.g. in terms of compensatory pathways) may be expected when comparing gene knockouts for 7 days (as done here) versus methods used by others?
- Hits and most validation experiments were performed using cells in which CD63 or CD9 were overexpressed, which does not reflect the physiological situation. Can the authors demonstrate that levels of CD63/CD9+ EVs released from wildtype cells (e.g. using Western Blotting or ELISA) are also affected by knockdown of the proteins or interference with the pathways discussed?
- Figure 2b. Why did the authors highlight ATPV1A and NDUFS1 here (proteins that were not followed up on)? Could the authors instead highlight the lower/upper hits from figure 3A?
- Figure 3a. Can the authors discuss why the highlighted proteins were chosen for validation in figure 3f, and not other lower/upper hits? Did the authors attempt to validate these other hits as well? Even if the authors could not validate them, this information would be valuable to include.
- Figure 3b/c. Did the authors test for potential effects of these compounds on cell growth/viability?
- Why did the authors perform a PS-capture ELISA to look for release of CD63/CD9+ EVs in figures 4D/E, and not use the CD63nLuc/CD9nLuc as in figure 5b? PS most likely captures only a subset of CD63/CD9+ EVs. The authors are therefore advised to repeat this experiment with the nLuc cells.
- Figure S2b. Can the authors explain the labeling of the blue line (plasmid library) and how this compares to the blue line in figure 1f (cells)? When they state "because the sEV barcoding efficacy was much lower" how do the authors define barcoding efficacy and where is this conclusion based on?
- Figure S15: It is well known that overexpression of CD63 or CD9 may affect their intracellular location. Therefore, analyzing CD63 and CD9 localization in CD63/CD9 CIBER cells (with overexpressed proteins) would be more informative.

Reviewer #2

(Remarks to the Author)

In their manuscript, the authors describe the development of a creative strategy for screening regulators of sEV biogenesis and release, called CIBER. As sEVs are secreted and ultimately uncoupled from their cell of origin, standard screening approaches are largely insufficient or not applicable. The authors address this gap with an elegant approach that relies on

barcoding the sEVs themselves to maintain the link between the sEV and the cell of origin. As a pooled screening format, CIBER provides a true high-throughput approach for identifying new factors and mechanisms in sEV biology. As highlighted by the authors, there are major bioengineering and therapeutic implications of better understanding and being able to directly modulate sEV production/release. In addition, the same conceptual approach could prove useful for screening and studying other types of secreted cellular products. Overall, the experiments are very well done, technically sound, with thorough consideration of potential pitfalls and means of controlling or accounting for any such caveats.

Main comments:

- In Fig 3a - Why did the authors set the z-RE thresholds for hits as +1.65 and -1.65? I see that they cited DeVore et al, but some more explanation about why they chose those thresholds would be helpful for understanding their rationale.
- In Fig 4a - For the comparison between the CD63 and CD9-CIBER screening results, have the authors considered comparing the conditions by looking at the values/genes that are some threshold distance from the diagonal (ie. one common threshold is 2 sd from the linear regression line), rather than those that lie within regions bounded by the 1.65/-1.65 thresholds? By only looking comparing values using these regions, there may be a tendency to over or under-estimate the degree of overlap between conditions. For example, values close to the corners of the regions are actually well correlated and may be less important to consider in the comparison between conditions.
- In Fig 4b - This is visually too complex to be able to connect the genes listed on the left with the GO terms listed on the right. I urge the authors to use an alternative way of visualizing this information that is more decipherable.
- In Fig 4c - For the highlighted GO terms, where do the associated genes lie in the CD63 vs CD9 z-RE plot? This would be helpful for determining to what degree they are unique to one condition vs the other.
- Fig 4d - For the rotenone inhibitor experiment, both CD63+ and CD9+ sEV release were reduced. This should be stated in the text (which currently only refers to the effect on CD63+ sEVs). While the effect does appear to be more pronounced for CD63+ sEVs, the effect for both sEV types appear to be significant. Accordingly, an additional statistical comparison should be done between vehicle and inhibition conditions within each sEV detection ab type.
- Supp Fig 15 - The cells shown have a very unusual morphology, even for HEK293T cells (they appear to be unusually 'triangular'). Do the authors have brightfield/phase-contrast or any additional fluorescent images they could provide from this experiment?
- Fig 5d - The flow plots should be shown as 2d histograms. The 3d representation makes comparison between conditions very difficult.

Minor comments:

- Fig 1e - If possible, the bioanalyzer plot x-axis should show fragment size rather than retention time, this will be more informative for gauging the actual size of the product.
- Supp Fig1 - "cell expansion" should read "cell expansion"
- Supp Fig5 - Although the results appear to be clear-cut, it would be informative to include a gate and the percentage of BFP+/- cells to confirm there is no major effect on Cas9 KO efficiency from co-expressing CD63-dCas9.
- Supp Fig 11 - Why does the node fill color range from -1.65 to 1.65? In the main figure (Fig 3h), these are the minimum z-RE values rather than the maximums.

Reviewer #3

(Remarks to the Author)

This manuscript by Koki Kunitake and colleagues describes a novel high-throughput screen that they developed to identify genes that are involved in the formation and release of a specific class of extracellular vesicles (EVs), that range in size between 30-200 nm and are generally referred to as small EVs (sEVs). There are multiple distinct classes of EVs that are produced by cells, and they all have been shown mediate intercellular communication, important for promoting several physiological processes and pathological conditions. However, the mechanisms that underlie their formation and release remain poorly understood and represents a pervasive question in the field.

Here, the authors generated a CRISPR based assay, which they name CRISPR-assisted individually barcoded sEV-based release regulator (CIBER), in order to identify genes important for the biogenesis of sEVs. This approach allowed for the detection of changes in the number of sub-classes of vesicles, i.e., CD-9 versus CD-63 expressing sEVs, generated by cells upon knocking-out a specific gene. The screen is innovative and was carefully performed, resulting in the identification of several genes/biological processes that were previously known to influence the formation and release of specific sub-classes of EVs, which the authors confirmed using siRNAs to knockdown a protein of interest or inhibitors to block the activity of a protein or biological process. Although these findings demonstrate that the screen is indeed working properly and represent an important start, ideally one would have liked to see the authors take the study a bit further to reveal novel mechanistic insights regarding how the shedding of CD-9 and CD-63 expressing sEVs is regulated or define biological contexts where this regulation is important. For example, the authors could determine whether cancer cells shed more CD-9 expressing sEVs due to their enhanced rates of growth, compared to their non-cancerous counterparts? If so, what regulators are uniquely expressed in those cells to promote the enhanced shedding of CD-9 expressing sEVs? Would blocking cell cycle progression in cancer cells similarly perturb the shedding of CD-9 expressing sEVs without affecting the production of CD-63 expressing sEVs? Alternatively, it would be interesting to demonstrate whether genes identified in the screen differentially regulate the formation and release of sub-classes of sEVs from cells following their treatment with a growth factor or a cellular stress. The addition of one or more of these experiments would significantly enhance the impact of the study. I also have a few additional points and concerns that should be addressed.

- 1) The screen only identifies proteins that are involved in EV biogenesis and shedding that are part of the vesicle cargo.

What about proteins that regulate EV formation/release but are not incorporated into EVs? This could be a potential limitation of the screen and should be discussed.

2) Along the same lines as the point above, the screen will only allow for the detection of changes in the number of EVs that express specific a marker, in this case CD-9 or CD-63. However, the issue is that there are likely sEVs that do not express either of these markers and thus regulators of these sEVs will not be identified in the screen. How do the author propose dealing with such an issue.

3) The findings showing the production of CD-9 expressing sEVs is correlated with cell cycle progression is an interesting observation, but ideally it should be developed more thoroughly. What is the mechanism responsible for this regulation? What gene identified in the screen mediates this effect, and how broadly relevant are these findings?

Version 1:

Reviewer comments:

Reviewer #1

(Remarks to the Author)

The authors have adequately addressed my previous concerns.

Reviewer #2

(Remarks to the Author)

The authors have done a very thorough job of addressing the reviewer comments and I have no further comments.

Reviewer #3

(Remarks to the Author)

I have carefully gone through the authors rebuttal to the points that I raised in the initial review of this manuscript. Overall, the authors have made a fair attempt to address those issues. While I feel that some of the underlying mechanisms responsible for how the shedding of the different classes of extracellular vesicles being examined is regulated might have been further developed, on balance, the authors have addressed the major points of my critique.

Point-by-point response to reviewers' comments for NCOMMS-23-54530-T, manuscript entitled "Barcoding of small extracellular vesicles with CRISPR-gRNA enables comprehensive, subpopulation-specific analysis of their biogenesis/release regulators"

RESPONSES TO REVIEWERS' COMMENTS

Reviewer #1 (Remarks to the Author):

The manuscript by Kunitake et al. describes a Cas9/gRNA-based barcoding system that allows a pooled screening approach to identify genes involved in EV release. Biogenesis of EVs remains incompletely understood, and this work offers an elegant, well-designed system to address this. The manuscript is generally well written, and its conclusions sufficiently supported by data. However, despite identification of novel sEV release regulators, the work remains largely descriptive, and identification of novel molecular mechanisms underlying EV release is lacking, which may somewhat limit the impact of this work. Furthermore, it contains several unclarities that need to be addressed.

We appreciate the reviewer's positive evaluation of our screening platform for sEV release regulators, and the valuable suggestions for improvement. Our point-by-point responses are given below. Furthermore, in the process of addressing points raised by other reviewers, we have demonstrated the portability of the CIBER screening platform to other cell lines and the generalizability of our findings about the effect of the cell cycle on sEV heterogeneity in cancer cells. We believe these additional results obtained during the revision process offer valuable insights that enhance the value of our study.

Specific comments:

1. EV isolation and characterization. The isolation procedure methods lack detail (e.g. g-forces used, UC tube types, rotor info). EV characterization is insufficient. The authors should, according to MISEV guidelines, at least characterize EV and potential contaminant marker expression compared to cell lysate expression (e.g. using Western Blot) and perform some type of EV imaging (e.g. EM).

A #1. We have conducted additional experiments to confirm the marker expression, purity and morphology of sEV obtained after ultracentrifugation with iodixanol by Western blotting and transmission electron microscopy (TEM) (new Supplementary Fig. 2). Western blots show the enrichment of two common sEV markers (native CD63 and CD9) and depletion of a negative marker (ER-bound calnexin, which is usually not present in sEVs), as well as the expression of dCas9-fused CD63/CD9, which we used for sEV barcoding. The TEM images show the typical cup-shaped appearance of sEVs; possible contaminants such as protein aggregates were not detected.

We believe these results, together with the nanoparticle tracking analysis data, show that the characterization of the sEVs used in this study meets the recommendations of MISEV2023, which is the most recent guideline for sEV studies according to the International Society of Extracellular Vesicles. We did find that the expression ratio of dCas9-fused CD63/CD9 to endogenous CD63/CD9 was lower in the sEV fraction than in the cellular fraction, which indicates limited loading of the fusion protein into sEVs. It might be worth addressing this point in a future study, but we do want to emphasize that the current sEV barcoding strategy already provides unbiased loading of barcode gRNA and is therefore suitable for reliable screening.

As for the information on sEV purification, the reviewer may have overlooked the part of the Methods section that already contained the requested information.

2. As discussed by the authors, many of the proteins previously identified to play roles in EV release (Rab27a, Alix, nSmase2, etc) were not picked up in the screen. This requires more attention and discussion. What effects of knockout of these proteins on EV release were found in the screen? What information has been shown by others for the cell line used in this work (HEK293T cells)? And what differences (e.g. in terms of compensatory pathways) may be expected when comparing gene knockouts for 7 days (as done here) versus methods used by others?

A #2. We thank the reviewer for these important points. Few studies have systematically addressed sEV release regulators in different cell lines (including HEK293T cells), and there may be cell-type-specific sEV release regulators (see also A #1 to reviewer 3), so it is not easy to comprehensively compare our results with those of other studies. In addition, CIBER screening contains both false positives and false negatives (see also A #5), as is inevitable in this kind of large-scale screening. Therefore, it is difficult to explain why some of the known sEV regulators have not been identified in CIBER screening. There seem to be many possible explanations, including the following.

Firstly, some Rabs families are reported to demonstrate functional redundancy, and it has been emphasized that careful analysis of Rab family genes is necessary in case of genome-wide KO screening (*J. Cell Biol.* 2019, 218, 2035). Also, in the same paper, it was reported that the KO of some Rab proteins gradually kills the cells, suggesting that the assay timeline affects the assay results.

To investigate this, we conducted additional experiments to address the effect of timeline in our case. We knocked out RAB27A and PDCD6IP (ALIX-encoding gene) with the gRNA present in the library used in our CIBER screening, and conducted RNA-seq at different time points (Days 3, 5 and 7; New Supplementary Fig. 28). We also tried a KO experiment for nSMase2, but we could not clearly confirm successful KO of nSMase2 at either the RNA or protein level, so we have not included the data. On the other hand, the RAB27A and PDCD6IP transcripts were reduced (new

Supplementary Fig. 28b). A Western blot showed complete removal of RAB27A at all time points, whereas ALIX (the gene product of PDCD6IP) was gradually reduced but still remained even at Day 7 (new Supplementary Fig. 28c; this is consistent with a report showing that the half-life of ALIX is relatively long (*Nature* 2011, 473, 337–342)). This observation suggests that Day 7 (when we replaced the culture media for sEVs collection) might be too early for analyzing the function of proteins with a long half-life, even though analysis at Day 7 is generally recommended for CRISPR screening (*Nat. Protoc.* 2017, 12, 828).

We also examined time-dependent changes in the expression of various gene sets upon KO of the above genes by means of GSEA Preranked analysis (new Supplementary Fig. 28e). Notably, gene sets whose expression was significantly and time-dependently changed were more concentrated in CIBER screening hits than in other non-CIBER-hit gene sets (new Supplementary Fig. 28f). This suggests that the KO of those genes alters the expression levels of other genes regulating sEV release, which may imply that some compensation occurred. Also, focusing on the individual CIBER-hit genes, the expression of multiple genes validated to affect sEV release seemed to change depending on the assay time point, which further suggests that the assay timeline might affect the results (new Supplementary Fig. 28g).

In summary, we think there could be many reasons why some of the known sEV regulators were not detected as hits. Changing the screening format to CRISPRa/i might be an option to partially solve this issue, because of the fast assay timeline as well as the different gRNAs used. It might be hard to address the redundancy issue with a normal CRISPR screening format. Applying combinatorial KO screening with focused target genes might also be an option (e.g. *STAR Protoc.* 2022, 3, 101556), but we feel this is beyond the scope of our present study because a new sEV-barcoding strategy would be necessary.

We also want to emphasize that the individual false positives and negatives in this study were found during the development of our high-throughput screening platform. Conducting CIBER screening under a variety of conditions and with multiple cell lines in future studies would be necessary to clarify this issue, and might also lead to the identification of other, previously unknown factors controlling sEV release.

Related discussion has been added in the Discussion section, and supplementary information (Discussion regarding Supplementary Fig. 28).

3. Hits and most validation experiments were performed using cells in which CD63 or CD9 were overexpressed, which does not reflect the physiological situation. Can the authors demonstrate that levels of CD63/CD9+ EVs released from wildtype cells (e.g. using Western Blotting or ELISA) are also affected by knockdown of the proteins or interference with the pathways discussed?

A #3. We consistently used CD63-nluc/CD9-nluc reporter assay, which we have rigorously confirmed the validity for sEV quantification in our previous study (Sci. Rep. 2018, 8, 14035), when relatively high-throughput and high sensitivity are necessary (i.e., many conditions, appropriate replicates, small scale, time-dependent measurement). However, as the reviewer points out, it is important to experimentally confirm applicability for native sEVs that do not overexpress CD63 and CD9. We therefore confirmed this as follows. (1) We examined the effect of inhibition of PI4KA and FASN on sEV release by nanoparticle tracking analysis with native HEK293T cells. The results are consistent with the finding that these two genes are release regulators of both CD63⁺ sEVs and CD9⁺ sEVs, since both of them are among the major sEV subpopulations (Fig. 1d,e). (2) We showed that mitochondria and lysosomes have different effects on the release of CD63⁺ sEVs and CD9⁺ sEVs. In this regard, we had conducted only phosphatidylserine capture ELISA for quantification in the initial submission, but that measurement might have involved some bias in the sEV population, as suggested by the reviewer. Therefore, we conducted additional experiments using CD63-CD63 or CD9-CD9 sandwich ELISA with sEVs released from normal HEK293T cells (new Supplementary Fig. 20a). We also conducted additional nluc-based reporter assay (new Supplementary Fig. 20b). All the results were consistent, providing further support for our conclusion that the effect of the pathway which we have found to affect the heterogeneity of sEVs with CIBER screening also affects the heterogeneity of native sEVs (please also see A #7).

4. Figure 2b. Why did the authors highlight ATPV1A and NDUFS1 here (proteins that were not followed up on)? Could the authors instead highlight the lower/upper hits from figure 3A?

A #4. We agree that it would be preferable to highlight genes whose functions have been experimentally addressed. Therefore, ATP6V1A has been replaced by PI4KA (note that ATP6V1A is a subunit of V-type ATPase and has already been confirmed to be a true upper hit by employing concanamycin A). As for NDUFS1, we have included the validation result of NDUFS1 by KD assay (new Fig. 3a).

5. Figure 3a. Can the authors discuss why the highlighted proteins were chosen for validation in figure 3f, and not other lower/upper hits? Did the authors attempt to validate these other hits as well? Even if the authors could not validate them, this information would be valuable to include.

A #5: Thank you. We have added new Supplementary Fig. 12 and a related note in the Supplementary Information to discuss this point. In addition to the data provided in the initial submission, we included data for other genes tested with CD63-CIBER. NDUFS1 and KIAA1109 were confirmed as additional validated hits (new Fig. 3f). As for COG3, COPG1, and VPS25, the results of the validation experiment proved variable, so we did not include the data and included the following comment in the note after Supplementary Fig. 12. “Knockdown of COG3, COPG1, and VPS25 showed a change of the nluc signal as predicted by the CIBER screening in one experiment, but this was not reproducible, so further validation will be needed (data not shown)” For your reference, the data is shown here.

As for YKT6, KD of this gene consistently increased sEV release, though it was identified as a lower hit. There are conflicting reports as to whether YKT6 enhances or suppresses sEV biogenesis (*elife* 7, e41460 (2018), *Nat. Cell Biol.* 14, 1036–1045 (2012), *Annu. Rev. Cell Dev. Biol.* 30, 255–289 (2014)). Our results might indicate that the effect on sEV release depends on the experimental setting, as was the case for ESCRT proteins. From this viewpoint, it might be worth checking the validity of the CIBER screening hit genes with high absolute values of z-REs (regardless of positive or negative values) depending on the experimental purpose. Related discussion has been added as a note after Supplementary Fig. 12.

Overall, 10 out of 14 candidate genes in CD63-CIBER screening with HEK293T cells were validated as true hits, which we believe is an acceptable proportion in pooled, large-scale screening (as discussed in the main text).

In the process of responding to other reviewers’ comments, we also conducted additional small-scale CD63-CIBER screening with a different cell line, SH-SY5Y (neuroblastoma-derived) (new Supplementary Fig. 15) (see also A #1 to reviewer 3). In this screening, 3 out of 3 candidate genes were confirmed to be true hits. However, we also noticed that KD of these genes in HEK293T cells suppressed the release of CD63⁺ sEVs, though the genes were not hits in CD63-CIBER screening in HEK293T cells (note that the degrees of suppression of sEV release between SH-SY5Y cells and 293T cells were different, as predicted from the screening results). This means that CIBER screening also produces a certain proportion of false negatives.

We do want to emphasize that false positives and negatives are inevitable in this kind of large-scale screening. Nevertheless, exploratory bioinformatic analyses with a low threshold are important to avoid missing significant biological pathways, and indeed this approach worked efficiently for finding biological pathways that differently affect the release of CD63⁺ sEVs and CD9⁺ sEVs (see also response to the comments by reviewer 2 (A #1 and A#4)). Furthermore, it is

also important to note that it is impossible to predict the hit accuracy of a comprehensive screening platform like CIBER screening before establishing the system. Conducting CIBER screening under a variety of conditions with multiple cell lines in the future studies would be necessary to address this issue, and would likely lead to the identification of other previously unknown factors (and gene networks) controlling sEV release. Related discussion has been added to the main text (line; 267 ~274) and supplementary information (note after Supplementary Fig. 15).

6. Figure 3b/c. Did the authors test for potential effects of these compounds on cell growth/viability?

A #6: In Fig. 3b/c, we showed the luminescence derived from CD63-nluc / CD9-nluc normalized by the protein amount of the sEV producer cells, and this reflects the cell number (viability). To clarify our assessment of the effects of these compounds on cell growth/viability, we also added the data before normalization in new Supplementary Fig. 9 so that readers can check both reporter CD63-nluc / CD9-nluc and cellular protein levels.

7. Why did the authors perform a PS-capture ELISA to look for release of CD63/CD9+ EVs in figures 4D/E, and not use the CD63nLuc/CD9nLuc as in figure 5b? PS most likely captures only a subset of CD63/CD9+ EVs. The authors are therefore advised to repeat this experiment with the nLuc cells.

A #7: Although we rigorously confirmed the validity of the CD63-nluc/CD9-nluc reporter assay for sEV quantification in our previous study (*Sci. Rep.* 2018, 8, 14035), we thought it would be preferable to show the data of non-labeled native sEVs where possible in this study, which is why we performed PS-capture ELISA in the original submission (Fig. 4d/e). PS has been shown to be exposed on a wide range of sEV populations and the ELISA based on PS capture allows for robust and sensitive detection of sEVs (*Sci. Rep.* 2016, 33935). However, we agree that PS capture focuses on a limited subset of sEVs, as the reviewer pointed out. In light of this, we have conducted additional experiments to support the data of Fig. 4 d/e with CD63-nluc/CD9-nluc reporter (new Supplementary Fig. 20b). Further, we also conducted CD63-CD63 or CD9-CD9 sandwich ELISA to assay native sEVs while excluding the unintended effect of PS-capture (new Supplementary Fig. 20a). The results of the PS-capture ELISA, nluc reporter assay, and sandwich ELISA were all consistent. This confirmation of our findings with complementary assays provides additional support for our conclusions in this study.

As for Fig. 5b, we chose reporter assay with CD63-nluc and CD9-nluc, since the amount of EVs released in the short period of time was very limited, so it was essential to use a sensitive assay platform. Given the consistency between the reporter assay and other assays with native sEVs (PS capture ELISA or sandwich ELISA) as well as our previous assessment of the reporter assay as

mentioned above, we believe our claims are well supported by the experimental data.

8. Figure S2b. Can the authors explain the labeling of the blue line (plasmid library) and how this compares to the blue line in figure 1f (cells)? When they state “because the sEV barcoding efficacy was much lower” how do the authors define barcoding efficacy and where is this conclusion based on?

A #8: Thank you for pointing out this. Based on the data of old Supplementary Fig. 2b, we concluded in the early stage of this study that the MS2-MCP-based barcoding system was not effective enough for screening. We assumed that the lentivirus production and infection step would not drastically skew the barcode composition because we followed the guideline for CRISPR screening (*Nat. Protoc.* 2017, 12, 828). However, it would have been preferable if we had obtained the barcode composition from the cellular fraction in this assay. We conducted additional experiments to re-test the performance of the MS2-MCP-based barcoding system, and again confirmed worse performance compared to the dCas9-based system (the barcoding rate in the re-test was actually worse than in the initial trial, leaving the conclusion unchanged). Now, in the new Supplementary Fig. 3b, the result of the initial trial is shown as expt.1 and the result of the new trial is shown as expt.2.

9. Figure S15: It is well known that overexpression of CD63 or CD9 may affect their intracellular location. Therefore, analyzing CD63 and CD9 localization in CD63/CD9 CIBER cells (with overexpressed proteins) would be more informative.

A #9: Thank you for the important suggestion. We conducted additional experiments to assess the intracellular localization of CD63-dCas9 and CD9-dCas9 by immunofluorescence (new Supplementary Fig. 29). As a result, while CD63-sfGFP and CD9-sfGFP showed distinctive endosomal localization and plasma membrane localization, respectively, as has been reported in other studies, we observed strong signals of CD63-dCas9 and CD9-dCas9 from ER-like inner membrane structure. This is probably due to slow intracellular trafficking of the protein, because the expression ratio of dCas9-fused CD63/CD9 to endogenous CD63/CD9 was lower in the sEV fraction than in the cellular fraction (see also #A1). On the other hand, we confirmed that a portion of the CD9-dCas9 certainly reaches the plasma membrane where native CD9 is dominantly localized, and also that a portion of CD63-dCas9 colocalizes with CD63-sfGFP, supporting the idea that CD63-dCas9 reaches endosomes where native CD63 is dominantly localized. Though it would be desirable in the future to develop a new gRNA loading strategy to achieve a localization of the gRNA recruiting protein that better reflects the physiological nature of the EV marker proteins, we wish to emphasize that the CIBER screening with the current fusion protein successfully identified

true sEV release regulators that consistently affect native sEVs. This observation supports the idea that dCas9 fusion proteins showing the correct intracellular localization are the major contributors to the screening results. Thus, we believe the major findings of this study as well as the validity of the concept of sEV barcoding with dCas9-fused sEV marker protein are not significantly affected by the slowness of CD63-dCas9 and CD9-dCas9 trafficking. Related discussion can be found in the discussion section in the main text and the legend of new Supplementary Fig. 29. We also commented on ways to address this issue in future studies, including the possibility of constructing knock-in cells to avoid overexpression of the fusion protein. In addition, we showed that the gRNA barcoding strategy works with other sEV marker proteins as well, including the soluble sEV protein ALIX, indicating that the use of different sEV marker proteins might also be an option (new Supplementary Fig. 31).

Reviewer #2 (Remarks to the Author):

In their manuscript, the authors describe the development of a creative strategy for screening regulators of sEV biogenesis and release, called CIBER. As sEVs are secreted and ultimately uncoupled from their cell of origin, standard screening approaches are largely insufficient or not applicable. The authors address this gap with an elegant approach that relies on barcoding the sEVs themselves to maintain the link between the sEV and the cell of origin. As a pooled screening format, CIBER provides a true high-throughput approach for identifying new factors and mechanisms in sEV biology. As highlighted by the authors, there are major bioengineering and therapeutic implications of better understanding and being able to directly modulate sEV production/release. In addition, the same conceptual approach could prove useful for screening and studying other types of secreted cellular products. Overall, the experiments are very well done, technically sound, with thorough consideration of potential pitfalls and means of controlling or accounting for any such caveats.

We are grateful for the reviewer's positive evaluation of our CIBER screening platform, and appreciate the valuable comments, which we have addressed as follows.

Main comments:

1. In Fig 3a - Why did the authors set the z-RE thresholds for hits as +1.65 and -1.65? I see that they cited DeVore et al, but some more explanation about why they chose those thresholds would be helpful for understanding their rationale.

A #1. If the histogram of z-RE of tested genes shows a normal distribution, ± 1.65 of z-RE corresponds to the 95th and 5th percentiles. We have now included this information for the sake of clarity in the main text. This cut-off value is relatively loose, but we did not want to discard true hits by using an overly strict cut-off. Indeed, our results support the idea that exploratory bioinformatic analysis with a low threshold is a powerful strategy; it enabled us to extract an important pathway involved in sEV release (see also A#5 to reviewer 1). Related discussion has been added in the discussion section in the main text.

2. In Fig 4a - For the comparison between the CD63 and CD9-CIBER screening results, have the authors considered comparing the conditions by looking at the values/genes that are some threshold distance from the diagonal (ie. one common threshold is 2 sd from the linear regression line), rather than those that lie within regions bounded by the 1.65/-1.65 thresholds? By only looking comparing values using these regions, there may be a tendency to over or under-estimate the degree of overlap between conditions. For example, values close to the corners of the regions are actually well correlated and may be less important

to consider in the comparison between conditions.

A #2. Fig. 4a is designed to broadly illustrate the overlap and difference of screening results of z-RE in CD63-CIBER and CD9-CIBER, and our detailed biological discussion (especially the difference between CD63-CIBER and CD9-CIBER) does not actually rely on Fig. 4a. We separately used all the z-REs of CD63-CIBER and CD9-CIBER screening for GO analysis and GSEAPreranked, and then compared the results. For this analysis, we did not select candidate genes in advance by using Fig. 4a, since we wished to retain as much information as possible for informatic analysis. However, we agree that separating genes by some threshold from the diagonal line would be more straightforward for visualization, so we have modified Fig. 4a by shading the region between ± 2 sd of the z-RE difference.

3. In Fig 4b - This is visually too complex to be able to connect the genes listed on the left with the GO terms listed on the right. I urge the authors to use an alternative way of visualizing this information that is more decipherable.

A #3. Thank you for the suggestion. We have revised Fig. 4b to improve the clarity by using a matrix-like visualization to show the relationship between GO terms and genes. GO terms grouped as oxidative phosphorylation are arrayed in a row with a heat map showing the adjusted p-value at the left of the figure. Genes annotated to the terms are arranged horizontally with the heat map showing z-RE at the top of the figure. If a gene is annotated to a term, a black point is placed at the intersection of the grids extending from the gene name and the term name.

4. In Fig 4c - For the highlighted GO terms, where do the associated genes lie in the CD63 vs CD9 z-RE plot? This would be helpful for determining to what degree they are unique to one condition vs the other.

A #4. Following the reviewer's advice, we have added new Supplementary Fig. 19 to show the location of associated genes. Along with the GSEAPreranked results, in CD63-CIBER compared to CD9-CIBER, more than half of genes (32/49) annotated to WP_MITOCHONDRIAL_COMPLEX_I_ASSEMBLY_MODEL_OXPPOS_SYSTEM have lower z-RE, and 19 out of 25 genes annotated to GOCC_PROTON_TRANSPORTING_V_TYPE_ATPASE_COMPLEX have higher z-RE. We note that most of the annotated genes are neither lower hits nor upper hits with the z-RE threshold, despite the significant difference of the gene set. This indicates that GSEAPreranked after CIBER screening is a powerful pipeline to extract biological pathways involved in sEV release processes, and works as a complementary approach to analysis of individual hit genes / GO analysis (note that GSEAPreranked does not require a cut-off

value to extract the hits beforehand, as mentioned in A#1).

5. Fig 4d - For the rotenone inhibitor experiment, both CD63⁺ and CD9⁺ sEV release were reduced. This should be stated in the text (which currently only refers to the effect on CD63⁺ sEVs). While the effect does appear to be more pronounced for CD63⁺ sEVs, the effect for both sEV types appear to be significant. Accordingly, an additional statistical comparison should be done between vehicle and inhibition conditions within each sEV detection ab type.

A #5. Firstly, in the process of addressing other reviewers' comments, we have conducted additional experiments. In addition to the PS-capture ELISA, we conducted CD63-CD63 or CD9-CD9 sandwich ELISA to exclude the possibility that we are focusing on a biased sEV population, and we also conducted nluc-based reporter assay (new Supplementary Fig. 20) (see also A #7 to reviewer 1). All the results are consistent and confirmed that the release of CD63⁺ sEVs is reduced more significantly than that of CD9⁺ sEVs, though the release of CD9⁺ sEVs was also reduced to some extent as pointed out by the reviewer. Actually, this is in line with our GSEA result showing that the gene set WP_MITOCHONDRIAL_COMPLEX_I_ASSEMBLY_MODEL_OXPPOS_SYSTEM has a significant negative enrichment score and that knockdown of NDUFS1, one of the major mitochondrial complex I subunits, reduces the release of not only CD63⁺ sEVs, but also CD9⁺ sEVs to some extent (new Supplementary Fig. 21). We should also point out that blocking mitochondrial function affects various biological pathways including the cell cycle (e.g., rotenone is a cell cycle inhibitor) and we have shown that halting the cell cycle reduces the release of CD9⁺ sEVs. So, it is natural that the release of CD9⁺ sEVs is affected by rotenone from this viewpoint. Still, we would like to emphasize that what we wanted to show was that we could identify pathways that act significantly differently on the release of different subpopulations of sEVs, and this purpose was achieved. We have added related discussion in the note after Supplementary Fig. 21.

6. Supp Fig 15 - The cells shown have a very unusual morphology, even for HEK293T cells (they appear to be unusually 'triangular'). Do the authors have brightfield/phase-contrast or any additional fluorescent images they could provide from this experiment?

A #6. Thank you for pointing this out. We have adopted another field of view for this figure with brightfield images (new Supplementary Fig. 22).

7. Fig 5d - The flow plots should be shown as 2d histograms. The 3d representation makes comparison between conditions very difficult.

A #7. We have revised Fig. 5d as requested.

Minor comments:

8. Fig 1e - If possible, the bioanalyzer plot x-axis should show fragment size rather than retention time, this will be more informative for gauging the actual size of the product.

A #8. We have revised the Fig. 1e in line with the reviewer's advice. It was not possible to show fragment size on the x-axis due to the Bioanalyzer setting. Instead, we have placed a gel image of a ladder with known fragment size next to the electropherogram.

9. Supp Fig1 - "cell expantion" should read "cell expansion"

A #9. Thank you. We have corrected the misspelling.

10. Supp Fig5 - Although the results appear to be clear-cut, it would be informative to include a gate and the percentage of BFP+/- cells to confirm there is no major effect on Cas9 KO efficiency from co-expressing CD63-dCas9.

A #10. We have revised the figure as requested (new Supplementary Fig. 6).

11. Supp Fig 11 - Why does the node fill color range from -1.65 to 1.65? In the main figure (Fig 3h), these are the minimum z-RE values rather than the maximums.

A #11. We have modified the color to cover a wider range of z-RE (new Supplementary Fig. 14).

Reviewer #3 (Remarks to the Author):

This manuscript by Koki Kunitake and colleagues describes a novel high-throughput screen that they developed to identify genes that are involved in the formation and release of a specific class of extracellular vesicles (EVs), that range in size between 30-200 nm and are generally referred to as small EVs (sEVs). There are multiple distinct classes of EVs that are produced by cells, and they all have been shown mediate intercellular communication, important for promoting several physiological processes and pathological conditions. However, the mechanisms that underlie their formation and release remain poorly understood and represents a pervasive question in the field.

Here, the authors generated a CRISPR based assay, which they name CRISPR-assisted individually barcoded sEV-based release regulator (CIBER), in order to identify genes important for the biogenesis of sEVs. This approach allowed for the detection of changes in the number of sub-classes of vesicles, i.e., CD-9 versus CD-63 expressing sEVs, generated by cells upon knocking-out a specific gene. The screen is innovative and was carefully performed, resulting in the identification of several genes/biological processes that were previously known to influence the formation and release of specific sub-classes of EVs, which the authors confirmed using siRNAs to knockdown a protein of interest or inhibitors to block the activity of a protein or biological process. Although these findings demonstrate that the screen is indeed working properly and represent an important start, ideally one would have liked to see the authors take the study a bit further to reveal novel mechanistic insights regarding how the shedding of CD-9 and CD-63 expressing sEVs is regulated or define biological contexts where this regulation is important.

We are grateful to the reviewer's positive evaluation on our screening platform for sEV release regulators. Our primary focus in this study is to establish the usefulness of gRNA barcoding for massively parallel identification of EV release regulators, and so addressing the detailed mechanism of each regulator is basically beyond the scope of this study. However, we agree that it would be preferable to take this study a bit further to show the generalizability of our concept and biological findings, and we have revised the manuscript in response to the reviewer's specific comments, as described below. We believe these changes help to clarify the value of this approach.

1. For example, the authors could determine whether cancer cells shed more CD-9 expressing sEVs due to their enhanced rates of growth, compared to their non-cancerous counterparts? If so, what regulators are uniquely expressed in those cells to promote the enhanced shedding of CD-9 expressing sEVs? Would blocking cell cycle progression in cancer cells similarly perturb the shedding of CD-9 expressing sEVs without affecting the production of CD-63 expressing sEVs? Alternatively, it would be interesting to demonstrate whether genes identified in the screen differentially regulate the formation and release of sub-classes of sEVs from cells following their treatment with a growth factor or a cellular stress. The

addition of one or more of these experiments would significantly enhance the impact of the study.

A #1. We appreciate the reviewer's insightful suggestions. We agree it is important to see how the shedding of CD-9 and CD-63 expressing sEVs is regulated and to define biological contexts where this regulation is important. From this viewpoint, we examined whether cell cycle arrest in cancer cells selectively perturbs the shedding of CD9⁺ sEVs and carried out dinaciclib assay with 2 other cancer cell lines, SH-SY5Y (neuroblastoma) and HT29 (colon cancer), for which growth and malignancy are reportedly related to sEV release processes (*Colloids Surf. A Physicochem. Eng. Asp.* 2017, 532, 195–202, *Am. J. Pathol.* 2017, 187, 1633–1647., etc.). Dinaciclib treatment suppressed the release of CD9⁺ sEVs, but not the release of CD63⁺ sEVs, in both SH-SY5Y and HT29 cells, suggesting that our findings are generalizable to other cell lines including cancer cells (new Fig. 5f). In the case of SH-SY5Y and HT29 cells, the release of CD63⁺ sEV was actually increased upon dinaciclib treatment, which suggests that there may be differences among cell types. Related discussion has been added in the discussion section.

As for the generalizability of our study, we also conducted a small-scale CD63-CIBER screening with the DTKP library (2,333 genes with 24,569 gRNAs, drug targets, kinases, and phosphatases) in SH-SY5Y cells to further examine the applicability of our system in other cell lines. The new screening was successful, supporting the portability of CIBER screening to other cell lines (new Supplementary Fig. 15). PI4KA was again detected as one of the top upper hits and validated to be a true hit, suggesting that this gene works as a potent sEV release regulator in multiple cell lines. We also tried to find cancer-specific lower hits, because it has been suggested that cancer-cell-specific inhibition of sEV release could be a therapeutic strategy for cancer. PKM and PGK1 were identified as candidate SH-SY5Y-specific lower hits. The knockdown of these genes indeed decreased CD63⁺ sEV release more significantly in SH-SY5Y cells than in HEK293T cells, showing the potential of CIBER screening to find cell-type-specific sEV release regulators. Interestingly, the cell-type specificity was not perfect, because moderate suppression of CD63⁺ sEV release was also observed in HEK293T cells. Genome-wide screening with a greater number of cell lines combined with robust bioinformatic analysis might be effective to find genes whose KD/KO shows large differences among different cell types, and this will be a focus of future work. Related discussion has been added to the main text (section entitled “Applicability of CIBER screening in multiple cell lines”) and the note after new Supplementary Fig. 15.

We believe these data are valuable additions to the manuscript, supporting the broad impact of our current work.

I also have a few additional points and concerns that should be addressed.

2. The screen only identifies proteins that are involved in EV biogenesis and shedding that are part of the vesicle cargo. What about proteins that regulate EV formation/release but are not incorporated into EVs? This could be a potential limitation of the screen and should be discussed.

A #2. CIBER screening can interrogate the effect of any gene on sEV biogenesis, so this appears to be a misunderstanding. We showed that many hit genes encode sEV-resident proteins registered in Vesiclepedia (Fig. 3g), which is an interesting finding that supports the idea that CIBER screening can identify genes related to sEVs, but it does not mean that only genes encoding sEV-resident proteins are explored. For example, the lower hit gene GOLGA2 encodes GM130, a Golgi-associated protein that is frequently used as a negative sEV marker to check for contamination of cellular components in isolated sEVs. To clarify this, we added a note in parenthesis saying: “note that this does NOT mean CIBER screening identifies only genes encoding sEV-resident proteins” (line 138)

3. Along the same lines as the point above, the screen will only allow for the detection of changes in the number of EVs that express specific a marker, in this case CD-9 or CD-63. However, the issue is that there are likely sEVs that do not express either of these markers and thus regulators of these sEVs will not be identified in the screen. How do the author propose dealing with such an issue.

A #3. In principle, our sEV barcoding strategy can be applied to a wide range of sEV subpopulations of interest, including those that do not express CD63 or CD9, simply by changing the protein used to recruit dCas9 into sEVs. In order to experimentally show this, we have conducted additional experiments to demonstrate that gRNA-barcoded sEVs can be constructed with dCas9 fused to other sEV markers such as ALIX (soluble), TSG101 (soluble) and PTGFRN (membrane bound) (new Supplementary Fig. 31). In addition, proteins modified with lipid tags efficiently recruit cytosolic protein into sEVs (*Nat. Commun.* 2024, 15, 5618), so it should be possible to use this kind of strategy to construct gRNA-barcoded sEVs without relying on sEV marker proteins such as tetraspanins. Relevant discussion has been added in the Discussion section.

4. The findings showing the production of CD-9 expressing sEVs is correlated with cell cycle progression is an interesting observation, but ideally it should be developed more thoroughly. What is the mechanism responsible for this regulation? What gene identified in the screen mediates this effect, and how broadly relevant are these findings?

A #4. Thank you for the comments. Regarding the broad relevancy, we conducted additional experiments to show the association of the cell cycle with the release of CD9⁺ sEVs in cancer cell lines (new Fig. 5f), see also A #1). We found that the release of CD9⁺ sEVs was consistently decreased upon cell cycle arrest, while release of CD63⁺ sEVs was not decreased, which at least supports the generality of this finding. Regarding the mechanism of this regulation, we checked the effect of the cluster of genes that are known to affect cellular viability (essential genes) on sEV release. We observed a clear tendency that KO of essential genes broadly reduces the release of CD9⁺ sEVs, rather than that of CD63⁺ sEVs (now added as new Supplementary Fig. 30). We think this observation suggests that the process of cell division itself, rather than some specific genes, might be driving the CD9⁺ sEV release process in synchrony with the cell cycle. Related discussion has been added to the discussion session.